# Differentiable and Stable Long-Range Tracking of Multiple Posterior Modes

**Ali Younis and Erik B. Sudderth**
{ayounis, sudderth}@uci.edu
Department of Computer Science, University of California, Irvine

## Abstract

Particle filters flexibly represent multiple posterior modes nonparametrically, via a collection of weighted samples, but have classically been applied to tracking problems with known dynamics and observation likelihoods. Such generative models may be inaccurate or unavailable for high-dimensional observations like images. We instead leverage training data to discriminatively learn particle-based representations of uncertainty in latent object states, conditioned on arbitrary observations via deep neural network encoders. While prior discriminative particle filters have used heuristic relaxations of discrete particle resampling, or biased learning by truncating gradients at resampling steps, we achieve unbiased and low-variance gradient estimates by representing posteriors as continuous mixture densities. Our theory and experiments expose dramatic failures of existing reparameterization-based estimators for mixture gradients, an issue we address via an importance-sampling gradient estimator. Unlike standard recurrent neural networks, our mixture density particle filter represents multimodal uncertainty in continuous latent states, improving accuracy and robustness. On a range of challenging tracking and robot localization problems, our approach achieves dramatic improvements in accuracy, while also showing much greater stability across multiple training runs.

## 1 Introduction

A *particle filter* (PF) [1–4] uses weighted samples to provide a flexible, nonparametric representation of uncertainty in continuous latent states. Classical PFs are generative models that typically require human experts to specify the latent state dynamics and measurement models. With the growing popularity of deep learning and abundance of time-series data, recent work has instead sought to discriminatively learn PF variants that (unlike conventional recurrent neural networks) capture uncertainty in latent states [5–8]. These approaches are especially promising for high-dimensional observations like images, where learning accurate generative models is extremely challenging.

The key challenge in learning discriminative PFs is the non-differentiable particle resampling step, which is key to robustly maintaining diverse particle representations, but inhibits end-to-end learning by preventing gradient propagation. Prior discriminative PFs have typically used heuristic relaxations of discrete particle resampling, or biased learning by truncating gradients. These methods severely compromise the effectiveness of PF training; prior work has often assumed exact dynamical models are known, or used very large numbers of particles for test data, due to these limitations.

After reviewing prior work on generative (Sec. 2) and discriminative (Sec. 3) training of PFs, Sec. 4 demonstrates dramatic failures of popular reparameterization-based gradient estimators for mixture models. This motivates our *importance weighted samples gradient* estimator, which provides the foundation for the *mixture density particle filter* of Sec. 5. Experiments in Sec. 6 show substantial advances over prior discriminative PF for several challenging tracking and visual localization tasks.

37th Conference on Neural Information Processing Systems (NeurIPS 2023).

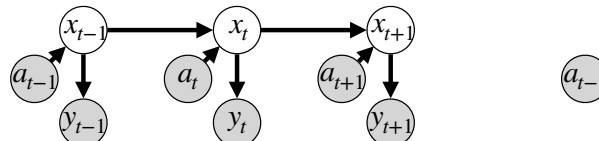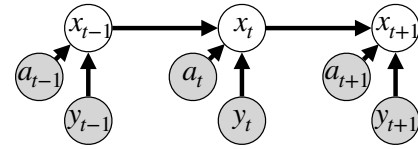

Figure 1: Sequential state estimation may be formulated via either generative (*left*) or discriminative (*right*) graphical models. In either case, dynamics of states $x_t$ are influenced by actions $a_t$, and estimated via data $y_t$.

## 2  Sequential State Estimation via Generative Models

Algorithms for sequential state estimation compute or approximate the posterior distribution of the system state $x_t$, given a sequence of observations $y_t$ and (optionally) input actions $a_t$, at discrete times $t = 1, \ldots, T$. Sequential state estimation is classically formulated as inference in a generative model like the *hidden Markov model* (HMM) or state space model of Fig. 1, with dynamics defined by state transition probabilities $p(x_t \mid x_{t-1}, a_t)$, and observations generated via likelihoods $p(y_t \mid x_t)$.

Given a known Markov prior on latent state sequences, and a corresponding observation sequence, the *filtered* state posterior $p(x_t \mid y_t, y_{t-1}, \ldots, y_1)$ can in principle be computed via Bayesian inference. When the means of state dynamics and observations likelihoods are linear functions, and noise is Gaussian, exact filtered state posteriors are Gaussian and may be efficiently computed via the *Kalman filter* (KF) [9–11]. The non-Gaussian state posteriors of general state space models may be approximated via Gaussians [12], but estimating the posterior mean and covariance is in general challenging, and cannot faithfully capture the multimodal posteriors produced by missing or ambiguous data.

### 2.1  Particle Filters

To flexibly approximate general state posteriors, particle filters (PF) [1–4] represent possible states nonparametrically via a collection of weighted samples or *particles*. The classical PF parameterizes the state posterior at time $t$ by a set of $N$ particles $x_t^{(:)} = \{x_t^{(1)}, \ldots, x_t^{(N)}\}$ with associated weights $w_t^{(:)} = \{w_t^{[1]}, \ldots, w_t^{(N)}\}$. PFs recursively update the state posterior given the latest observation $y_t$ and action $a_t$, and may be flexibly applied to a broad range of models; they only assume that the state dynamics may be simulated, and the observation likelihood may be evaluated.

**Particle Proposal.** To produce particle locations $x_t^{(:)}$ at time $t$, a model of the state transition dynamics is applied individually to each particle $x_{t-1}^{(i)}$, conditioned on external actions $a_t$ if available:

$$x_t^{(i)} \sim p(x_t \mid x_{t-1} = x_{t-1}^{(i)}, a_t). \tag{1}$$

**Measurement Update.** After a new particle set $x_t^{(:)}$ has been proposed, particle weights must be updated to account for the latest observation $y_t$ via the known likelihood function:

$$w_t^{(i)} \propto p(y_t \mid x_t^{(i)}) w_{t-1}^{(i)}. \tag{2}$$

The updated weights are then normalized so that $\sum_{i=1}^{N} w_t^{(i)} = 1$. This weight update is motivated by importance sampling principles, because (asymptotically, as $N \to \infty$) it provides an unbiased approximation of the true state posterior $p(x_t \mid y_t, \ldots, y_1) \propto p(y_t \mid x_t) p(x_t \mid y_{t-1}, \ldots, y_1)$.

**Particle Resampling.** Due to the stochastic nature of PFs, over time particles will slowly diverge to regions with low posterior probability, and will thus be given little weight in the measurement step. If all but a few particles have negligible weight, the diversity and effective representational power of the particle set is diminished, resulting in poor estimates of the state posterior.

To address this, particle resampling is used to maintain diversity of the particle set over time, and may be performed at every iteration or more selectively when the "effective" sample size becomes too small [13]. During resampling, a new uniformly-weighted particle set $\hat{x}_t^{(:)}$ is constructed by discrete resampling with replacement. Each resampled particle is a copy of some existing particle, where copies are sampled with probability proportional to the particle weights:

$$\hat{x}_t^{(i)} = x_t^{(j)}, \qquad j \sim \text{Cat}(w_t^{(1)}, \ldots, w_t^{(N)}). \tag{3}$$

After resampling, assigning uniform weights $\hat{w}_t^{(i)} = \frac{1}{N}$ maintains an unbiased approximate state posterior. This discrete resampling is non-differentiable, preventing gradient estimation via standard reparameterization [14–16], and inhibiting end-to-end learning of dynamics and measurement models.

## 2.2 Regularized Particle Filters

The regularized PF [17, 18] acknowledges that a small set of discrete samples will never *exactly* align with the true continuous state, and thus estimates a continuous state density $m(x_t \mid x_t^{(:)}, w_t^{(:)}, \beta)$ by convolving particles with a continuous kernel function $K$ with bandwidth $\beta$:

$$m(x_t \mid x_t^{(:)}, w_t^{(:)}, \beta) = \sum_{i=1}^{N} w_t^{(i)} K(x_t - x_t^{(i)}; \beta). \tag{4}$$

The kernel function could be a Gaussian, in which case $\beta$ is a (dimension-specific) standard deviation. The extensive literature on *kernel density estimation* (KDE) [19] provides theoretical guidance on the choice of kernel. Quadratic Epanechnikov kernels take $K(u; \beta) = \frac{3}{4} \left(1 - (u/\beta)^2\right)$ if $|u/\beta| \leq 1$, $K(u; \beta) = 0$ otherwise, and asymptotically minimize the mean-squared-error in the estimation of the underlying continuous density. A number of methods have been proposed for selecting the smoothing bandwidth $\beta$ [19–21], but they often have asymptotic justifications, and can be unreliable for small $N$.

Attractively, regularized PFs resample particles $\hat{x}_t^{(i)} \sim m(x_t \mid x_t^{(:)}, w_t^{(:)}, \beta)$ from the continuous mixture density rather than the discrete particle set. By resampling from a continuous distribution, regularized PFs ensure that no duplicates exist in the resampled particle set, increasing diversity while still concentrating particles in regions of the state space with high posterior probability. Our proposed *mixture density PF* generalizes regularized PFs, using bandwidths $\beta$ that are tuned jointly with discriminative models of the state dynamics and observation likelihoods.

# 3 Conditional State Estimation via Discriminative Particle Filters

While classical inference algorithms like KFs and PFs are effective in many domains, they require faithful generative models. A human expert must typically design most aspects of the state dynamics and observation likelihoods, and algorithms that use PFs to learn generative models often struggle to scale beyond low-dimensional, parametric models [22]. For complex systems with high-dimensional observations like images, learning generative models of the observation likelihoods is often impractical, and arguably more challenging than estimating latent states given an observed data sequence. We instead learn *discriminative* models of the distribution of the state *conditioned* on observations (see Fig. 1), and replace manually engineered generative models with deep neural networks learned from training sequences with (potentially sparse) latent state observations.

**Discriminative Kalman Filters.** Discriminative modeling has been used to learn conditional variants of the KF [23, 24] where the posterior is parameterized by a Gaussian state space model, and state transition and observation emission models are produced from data by trained neural networks. Like *conditional random field* (CRF) [25] models for discrete data, discriminative KFs do not learn likelihoods of observations, but instead condition on them. Discriminative KFs desirably learn a posterior covariance, and thus do not simply output a state prediction like conventional recurrent neural networks. But, they are limited to unimodal, Gaussian approximations of state uncertainty.

**Discriminative Particle Filters.** It is attractive to integrate similar discriminative learning principles with PFs, but the technical challenges are substantially greater due to the nonparametric particle representation of posterior uncertainty, and the need for particle resampling to maintain diversity.

Like in generative PFs, discriminative PFs update the particle locations using a *dynamics model* as in Eq. (1). Unlike classic PFs, the particle weights are not updated using a generative likelihood function. Instead, discriminative PFs compute new weights $w_t^{(:)}$ using a *measurement model* function $\ell(x_t; y_t)$ trained to properly account for uncertainty in the discriminative particle posterior:

$$w_t^{(i)} \propto \ell(x_t^{(i)}; y_t) w_{t-1}^{(i)}. \tag{5}$$

Here $\ell(x_t; y_t)$ is a (differentiable) function optimized to improve the accuracy of the discriminative particle posterior, rather than a generative likelihood.

Creating a learnable discriminative PF requires parameterizing the dynamics and measurement models as differentiable functions, such as deep neural networks. This is straightforward for the measurement model $\ell(x_t^{(i)}; y_t)$, which may be defined via any feed-forward neural network architecture like those typically used for classification (see Fig. 2). For the dynamics model, the neural network does not simply need to score particles; it must be used for stochastic simulation. Using reparameterization [14–16], dynamics simulation is decomposed as sampling from a standard Gaussian distribution, and then

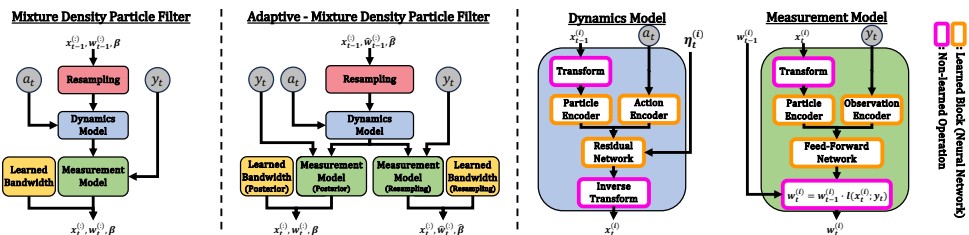

Figure 2: *Left:* Our MDPF method showing the various sub-components. *Middle:* Our A-MDPF method with decoupled measurement models and bandwidths. *Right:* Dynamics and measurement model structures used in MDPF and A-MDPF. The dynamics and measurement models are composed of several neural networks as well as some fixed transforms, which convert the angular state dimensions of particles into a vector representation.

transforming that sample using a learned neural network, possibly conditioned on action $a_t$:

$$x_t^{(i)} = f(\eta_t^{(i)}; x_{t-1}^{(i)}, a_t), \qquad \eta_t^{(i)} \sim N(0, I). \qquad (6)$$

The dynamics model $f(\eta; x_{t-1}, a_t)$ is a feed-forward neural network that deterministically processes noise $\eta$, conditioned on $x_{t-1}$ and $a_t$, to implement the dynamical sampling of Eq. (1). The neural network $f$ may be flexibly parameterized because discriminative particle filters only require simulation of the dynamical model, not explicit evaluation of the implied conditional state density.

### 3.1 Prior Work on Discriminative Particle Filters

Specialized, heuristic variants of discriminative PFs have been previously used for tasks like robot localization [26] and multi-object tracking [27]. Principled end-to-end learning of a discriminative PF requires propagating gradients through the PF algorithm, including the discrete resampling step. Zhu et al. [28] explore replacing the PF resampling step with a learned (deterministic) particle transform, but find that exploding gradient magnitudes prohibit end-to-end training.

**Truncated-Gradient Particle Filter (TG-PF).** The first so-called "differentiable" particle filter [5] actually treated the particle resampling step as a non-differentiable discrete resampling operation, and simply truncated all gradients to zero when backpropagating through this resampling. Though simple, this approach leads to biased gradients, and often produces ineffective models because *back-propagation through time* (BPTT [29]) is not possible. Perhaps due to these limitations, experiments in Jonschkowski et al. [5] assumed a simplified training scenario where the ground-truth dynamics are known, and only the measurement model must be learned.

**Discrete Importance Sampling Particle Filter (DIS-PF).** Ścibior et al. [6] propose a gradient estimator for generative PFs that may also be adapted to discriminative PFs. They develop their estimator by invoking prior work on score-function gradient estimators [30], but we provide a simpler derivation via importance sampling principles. Instead of discretely resampling particles as in Eq. (3), a separate set of weights $v_t^{(:)}$ is defined and used to generate samples:

$$\hat{x}_t^{(i)} = x_t^{(j)}, \qquad j \sim \text{Cat}(v_t^{(1)}, \ldots, v_t^{(N)}). \qquad (7)$$

To account for discrepancies between these resampling weights $v^{(:)}$ and true weights $w^{(:)}$, the resampled particle weights $\hat{w}_t^{(i)}$ are defined via importance sampling. The choice of $v_t^{(:)}$ critically impacts the effectiveness of the resampling step. To maintain the standard PF update of Eq. (3), we set $v_t^{(i)} = w_t^{(i)}$, yielding the following resampled particle weights and associated gradients:

$$\hat{w}_t^{(i)} = \frac{w_t^{(i)}}{v_t^{(i)}\big|_{v_t^{(i)} = w_t^{(i)}}} = 1, \qquad \nabla_\phi \hat{w}_t^{(i)} = \frac{\nabla_\phi w_t^{(i)}}{v_t^{(i)}\big|_{v_t^{(i)} = w_t^{(i)}}}. \qquad (8)$$

Intuitively, particle locations are resampled according to the current weights in the forward pass of DIS-PF. Then when computing gradients, perturbations of the observation and dynamics models are accounted for by changes in the associated particle weights.

A drawback of this discrete importance sampling is known as the *ancestor problem*: since the resampled particle set is constructed from a subset of the pre-resampled particles, no direct gradients exist for particles that were not chosen during resampling; they are only indirectly influenced by the weight normalization step. This increases the DIS gradient estimator variance, slowing training.

**Soft Resampling Particle Filter (SR-PF).** Karkus et al. [7] propose a *soft-resampling* (SR) mixing of the particle weights with a discrete uniform distribution before resampling: $v_t^{(i)} = (1-\lambda)w_t^{(i)} + \lambda/N$.

Viewing particle locations as fixed, SR-PF evaluates the resampled particle weights and gradients as

$$\hat{w}_t^{(i)} = \frac{w_t^{(i)}}{(1-\lambda)w_t^{(i)} + \lambda/N}, \qquad \nabla_\phi \hat{w}_t^{(i)} = \nabla_\phi \left( \frac{w_t^{(i)}}{(1-\lambda)w_t^{(i)} + \lambda/N} \right). \qquad (9)$$

While the SR-PF weight update is differentiable, it does not avoid the ancestor problem. By resampling low-weight particles more frequently, SR-PF will degrade overall PF performance when $N$ is not large. More subtly, the gradient of Eq. (9) has (potentially substantial) bias: it assumes that perturbations of model parameters influence particle weights, but *not* the outcome of discrete resampling (7). Indeed in some experiments in [7], smoothing is actually disabled by setting $\lambda = 0$.

**Concrete Particle Filter (C-PF).** The Gumbel-softmax or Concrete distribution [31, 32] approximates discrete sampling by interpolation with a continuous distribution, enabling reparameterized gradient estimation. Each resampled particle is a convex combination of the weighted particle set:

$$\hat{x}_t^{(i)} = \sum_{j=1}^N \alpha_{ij} x_t^{(j)}, \qquad \alpha_{ij} = \frac{\exp((\log(w_t^{(j)}) + G_{ij})/\lambda)}{\sum_{k=1}^N \exp((\log(w_t^{(k)}) + G_{ik})/\lambda)}, \qquad G_{ij} \sim \text{Gumbel}. \quad (10)$$

Gradients of (10) are biased with respect to discrete resampling due to the non-learned "temperature" hyperparameter $\lambda > 0$. When $\lambda$ is large, the highly-biased relaxation will interpolate between modes, and produce many low-probability particles. Bias decreases as $\lambda \to 0$, but in this limit the variance is huge. Even with careful tuning of $\lambda$, Concrete relaxations are most effective for small $N$. Maddison et al. [32] focus on models with binary latent variables, and find that performance degrades even for $N = 8$, far smaller than the $N$ needed for practical PFs. While we provide C-PF as a baseline, we are unaware of prior work successfully incorporating Concrete relaxations in PFs.

**Optimal Transport Particle Filter (OT-PF).** OT-PF [8] modifies PFs by replacing stochastic resampling with the optimization-based solution of an entropy-regularized *optimal transport* (OT) problem. OT seeks a probabilistic correspondence between particles $x_t^{(:)}$ with weights $w_t^{(:)}$ as in (5), and a corresponding set of uniformly weighted particles, by minimizing a Wasserstein metric $\mathcal{W}_{2,\lambda}^2$:

$$\min_{\tilde{\alpha} \in [0,1]^{N \times N}} \sum_{i,j=1}^N \tilde{\alpha}_{ij} \left( ||x_t^{(i)} - x_t^{(j)}||^2 + \lambda \log \frac{\tilde{\alpha}_{ij}}{N^{-1} w_t^{(j)}} \right), \text{ s.t. } \sum_{j=1}^N \tilde{\alpha}_{ij} = \frac{1}{N}, \sum_{i=1}^N \tilde{\alpha}_{ij} = w_t^{(j)}. \quad (11)$$

Entropy regularization is required for differentiability, and OT-PF accuracy is sensitive to the non-learned hyperparameter $\lambda > 0$. OT-PF uses this entropy-regularized mapping to interpolate between particles: $\hat{x}_t^{(i)} = \sum_{j=1}^N N\tilde{\alpha}_{ij} x_t^{(j)}$. This assignment approximates the results of true stochastic resampling in the limit as $N \to \infty$, but lacks accuracy guarantees for moderate $N$.

OT-PF solves a *regularized* OT problem which relaxes discrete resampling, producing biased gradients that are reminiscent (but distinct) from C-PF. Minimization of Eq. (11) via the Sinkhorn algorithm [33] requires $\mathcal{O}(N^2)$ operations. This OT problem must be solved at each step of both training and test, and in practice, OT-PF is substantially slower than all competing discriminative PFs (see Fig. 10).

## 4 Stable Gradient Estimation for Mixture Model Resampling

Training our discriminative *mixture density particle filter* (MDPF) requires unbiased and low-variance estimates of gradients of samples from continuous mixtures. Mixture sampling is classically decomposed into discrete (selecting a mixture component) and continuous (sampling from that component) steps. The Gumbel-softmax or Concrete distribution [31, 32] provides a reparameterizable relaxation of discrete resampling, but may have substantial bias. Some work has also applied OT methods for gradient estimation in restricted families of Gaussian mixtures [35, 36]. We instead develop importance-sampling estimators that are unbiased, computationally efficient, and low variance.

**Implicit Reparameterization Gradients.** Direct reparameterization of samples, as used for the latent Gaussian distributions in variational autoencoders [14–16], cannot be easily applied to mixture models. Reparameterized sampling requires an invertible standardization function $S_\phi(z) = \epsilon$ that transforms a sample $z$, drawn from a distribution parameterized by $\phi$, to an auxiliary variable $\epsilon$ independent of $\phi$. For mixture distributions, the inverse of $S_\phi(z)$ cannot be easily computed.

The *implicit reparameterization gradients* (IRG) estimator [37, 38] avoids explicit inversion of the standardization function $S_\phi(z)$. Using implicit differentiation, reparameterization can be achieved so long as a computable (analytically or numerically) and invertible $S_\phi(z)$ exists, without the need to

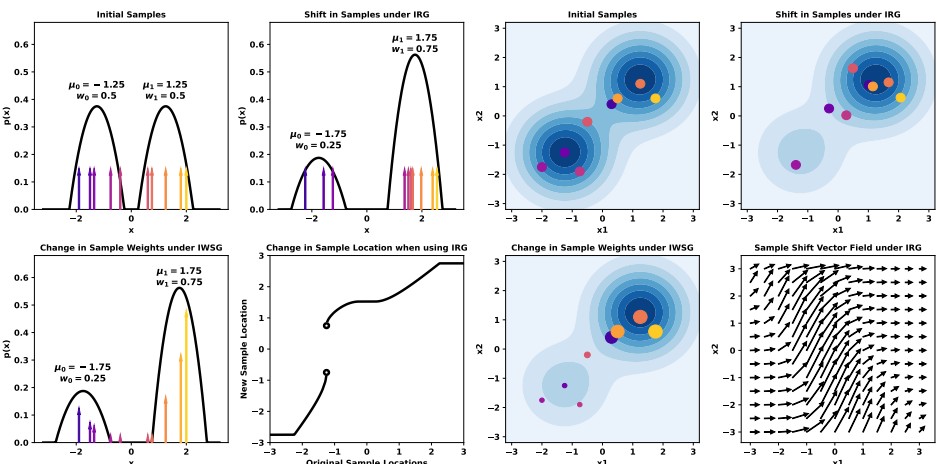

Figure 3: *Left:* Example changes in samples from a 1D mixture model with two Epanechnikov [34] components. Under IRG, changes in the mixture are shown by dramatic sample shifting, where some samples must change modes to account for the changes in the relative weights of each mixture mode. Explicitly plotting the particle transformation induced by IRG reveals a discontinuity (bottom). In contrast, IWSG smoothly reweights samples. *Right:* Example changes in samples from a 2D mixture of two Gaussians. IRG again induces large shifts as particles change modes, as demonstrated by the vector field (bottom).

explicitly compute its inverse. IRG gradients are expressed via the Jacobian of $S_\phi(z)$:

$$\nabla_\phi z = -(\nabla_z S_\phi(z))^{-1} \nabla_\phi S_\phi(z). \tag{12}$$

For univariate distributions, the cumulative distribution function (CDF) $M_\phi(z)$ is a valid standardization function. For higher dimensions, the multivariate distributional transform [39] is used:

$$S_\phi(z) = \Big( M_\phi(z_1), M_\phi(z_2 \mid z_1), \dots, M_\phi(z_D \mid z_1, \dots, z_{D-1}) \Big). \tag{13}$$

While IRG may be applied to any distribution with continuous CDF, and has been explicitly suggested (without experimental validation) for mixtures [37], we show that it may have enormous variance.

**Importance Weighted Sample Gradient Estimator.** We propose a novel alternative method for computing gradients of samples from a continuous mixture distribution. Our *importance weighted sample gradient* (IWSG) estimator employs importance sampling for unbiased gradient estimation. IWSG is related to the DIS-PF gradient estimator for discrete resampling [6], but we instead consider continuous mixture distributions with arbitrary component distributions. This resolves the ancestry problem present in Ścibior et al. [6], since particle locations could be sampled from multiple overlapping mixture components, whose parameters will all have non-zero gradients.

Formally, we wish to draw samples $z^{(i)} \sim m(z \mid \phi)$ from a mixture with parameters $\phi$. Instead of sampling from $m(z \mid \phi)$ directly, IWSG samples from some proposal distribution $z^{(i)} \sim q(z)$. Importance weights $w^{(i)}$, and associated gradients, for these samples then equal

$$w^{(i)} = \frac{m(z^{(i)} \mid \phi)}{q(z^{(i)})}, \qquad \nabla_\phi w^{(i)} = \frac{\nabla_\phi m(z^{(i)} \mid \phi)}{q(z^{(i)})}. \tag{14}$$

Setting the proposal distribution $q(z) = m(z \mid \phi_0)$ to be a mixture model with the "current" parameters $\phi_0$ at which gradients are being evaluated, the IWSG gradient estimator (14) becomes

$$w^{(i)} = \frac{m(z^{(i)} \mid \phi)\big|_{\phi=\phi_0}}{m(z^{(i)} \mid \phi_0)} = 1, \qquad \nabla_\phi w^{(i)} = \frac{\nabla_\phi m(z^{(i)} \mid \phi)\big|_{\phi=\phi_0}}{m(z^{(i)} \mid \phi_0)}. \tag{15}$$

While current samples are given weight one because they are exactly sampled from a mixture with the current parameters $\phi_0$, gradients account for how importance weights will change as mixture parameters $\phi$ deviate from $\phi_0$. Note that gradients are *not* taken with respect to the proposal $q(z) = m(z \mid \phi_0)$ in the denominator of $w^{(i)}$, since sample locations $z^{(i)}$ are not altered by gradient updates; changes in the associated importance weights are sufficient for unbiased gradient estimation.

**Instability of Implicit Reparameterization Gradients.** Reparameterization methods capture changes to a given distribution $p(z|\phi)$, and thus to its parameters $\phi$, by shifting the samples drawn from that distribution. When $p(z|\phi)$ is unimodal, this induces a smooth shift in the samples. When $p(z|\phi)$ contains multiple non-overlapping modes, these shifts are no longer smooth as samples

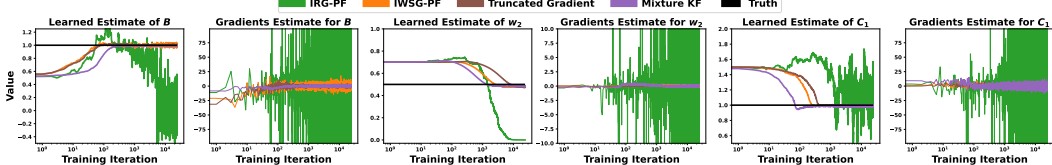

Figure 4: A simple temporal prediction problem where IRG has highly unstable gradient estimates. We use $N = 25$ particles when training IRG-PF, IWSG-PF, and truncated gradients (TG-PF). We show the learned values for parameters $B, w_2, C_1$ and their gradients during training. IRG produces unstable gradients which prevent parameters from converging at all, but IWSG allows for smooth convergence of all parameters. IWSG is faster than biased TG, and nearly as effective as (expensive, and for general models intractable) mixture KF.

jump from one mode to another; see Fig. 3. For mixtures with finitely supported kernels (like the Epanechnikov) and non-overlapping modes, discontinuities exist in $S_\phi^{-1}(\epsilon)$, and IRG estimates may have infinite variance. These discontinuities correspond to samples shifting from one mode to another, and the resulting non-smooth sample perturbations will destabilize backpropagation algorithms for gradient estimation. While the inverse CDF is always continuous for Gaussian mixtures, it may still have near-infinite slope when modes are widely separated, and induce similar IRG instabilities.

In contrast, our IWSG estimator reweights particles instead of shifting them. This reweighting is smooth and does not suffer from any discontinuities. To illustrate this, we create a simple discriminative PF with linear-Gaussian dynamics (16) and a bimodal observation likelihood (17):

$$p(x_t \mid x_{t-1}, a_t) = \text{Norm}(x_t \mid Ax_{t-1} + Ba_t, \sigma^2), \tag{16}$$

$$p(y_t \mid x_t) = w_1 \text{Norm}(y_t \mid C_1 x_t + c_1, \gamma^2) + w_2 \text{Norm}(y_t \mid C_2 x_t + c_2, \gamma^2). \tag{17}$$

Exact posterior inference for this model is possible via a mixture KF [11, 40] which represents the state posterior as a Gaussian mixture. The number of mixture components grows exponentially with time, motivating PFs for long-term tracking, but remains tractable over a few time steps.

Using a fixed dataset and gradient descent, we train discriminative PFs using the IRG estimator, our IWSG estimator, as well as with biased gradients that are truncated at each resampling step [5]. We initialize with perturbed parameter values and aim to learn the values of all dynamics and likelihood parameters, except for $\sigma$ and $\gamma$ which are fixed to their true values. See Appendix for details.

Fig. 4 shows the estimates and gradients for a subset of the learned parameters, and several gradient estimators. IRG's unstable gradients prevent convergence of all parameters. For training (generative) marginal PFs [41] which approximate marginals with mixtures, Lai et al. [42] found that IRG gradient variance was so large that biased estimators converged faster, but provided no detailed analysis. In contrast, IWSG converges smoothly for all parameters, and more rapidly than truncated gradients.

## 5 Mixture Density Particle Filters

Our *mixture density particle filter* (MDPF) is a discriminative model inspired by regularized PFs, where resampling uses KDEs centered on the current particles. We apply IWSG to this mixture resampling, achieving unbiased and low-variance gradient estimates for our fully-differentiable PF.

MDPF does not incorporate human-specified dynamics or measurement models; some prior work assumed known dynamics to simplify learning [5, 7]. These models are parameterized as deep neural networks (NNs), and trained via stochastic gradient descent to minimize the negative-log-likelihood of (potentially sparsely labeled) states $x_t$, given observations $y_t$ and (optionally) actions $a_t$. As shown in Fig. 2, dynamics and measurement models are flexible composed from smaller NNs. Our MDPF places no constraints on the functional form of NNs, allowing for modern architectures such as CNNs [43] and Spatial Transformers [44] to be used. See Appendix for implementation details.

The MDPF uses KDE mixture distributions as in Eq. (4) to define state posteriors used to evaluate the training loss, as well as resampling. A bandwidth parameter $\beta$ is thus required for each state dimension. Rather than setting $\beta$ via the classic heuristics discussed in Sec. 2.2, we make $\beta$ a learnable parameter, optimizing it using end-to-end learning along with the dynamics and measurement models. This contrasts with prior work including SR-PF [7] and C-PF and OT-PF [8], which all include relaxation hyperparameters that must be tuned via multiple (potentially expensive) training trials.

We also extend the MDPF by decoupling the mixtures used for particle resampling and posterior state estimation. Using two separate measurement models, we compute two sets of particle weights $\tilde{w}_t^{(:)}$

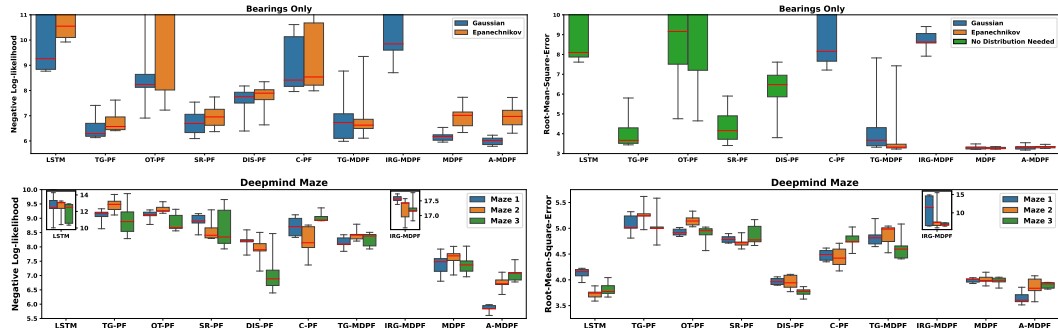

Figure 5: Box plots showing median (red line), inter-quartile (colored box) and range (whiskers) over several training runs on the Bearings-Only (11 runs) and Deepmind-Maze (5 runs) tracking tasks. MDPF and A-MDPF consistently perform well on both the NLL and RMSE metrics. Our IWSG estimator is critical: IRG-MDPF performs very poorly due to unstable gradients, TG-MDPF is inconsistent and sometimes becomes trapped in local optima, and baselines with biased gradient estimators typically have inferior performance. LSTM achieves low RMSE in the Deepmind-Maze task however we find it simply blind-propagates using the noisy actions. DIS-PF performs well for Maze 3 but with larger variability which can sometimes lead to worse performance than our more stable A-MDPF. Note that IRG-MDPF does not support Epanechnikov kernels, which induce discontinuous CDFs.

and $w_t^{(:)}$, one for resampling and the other for posterior state estimation; see Fig. 2. Each distribution is also given a separate bandwidth $\tilde{\beta}, \beta$. This *adaptive* mixture density PF (A-MDPF) allows the uncertainty in the estimation of the current latent state, and the degree of exploration in the particle resampling step, to be decoupled and separately optimized during training. Note that training of separate posterior and resampling distributions is *impossible* for PFs that truncate temporal gradients; it is only feasible due to our unbiased and low-variance IWSG estimator of resampling gradients.

## 6 Experiments

We evaluate our MDPF and A-MDPF on a variety state estimation tasks in complex, but simulated, environments. We compare to TG-PF [5], SR-PF [7], OT-PF [8], DIS-PF [6], as well as C-PF which uses the Concrete distribution [32, 31] to relax discrete resampling. We also compare to variants of MDPF: Truncated-Gradient-MDPF (TG-MDPF) is similar but stops gradients at each particle resampling, and IRG-MDPF replaces our IWSG estimator with the unstable IRG estimator. Finally, we compare to a LSTM [45] which produces point predictions via uninterpretable latent states.

For all tasks, we estimate posterior distributions of a 3D (translation and angle) state of a robot, $x_t = (x, y, \theta)$. We use Gaussian kernels for position components, and von Mises kernels for angular components, of all KDEs. As is common in *recurrent neural network* (RNN) training, we employ conservative gradient clipping [46], as well as truncated-BPTT [47] where gradients are propagated for 4 time-steps. All methods are initialized as the true state with added noise.

We train and evaluate MDPFs to minimize the following *negative-log-likelihood* (NLL) loss:

$$\mathcal{L}_{\text{NLL}} = \frac{1}{|\mathcal{T}|} \sum_{t \in \mathcal{T}} -\log(m(x_t | x_t^{(:)}, w_t^{(:)}, \beta)). \tag{18}$$

We use sparsely labeled training data to highlight the need for effective multi-time-step gradients during training. Dense labeling of data is often expensive, so many real-world datasets have similarly sparse labels. During training, we label true states every 4th time-step ($\mathcal{T} = \{4, 8, 12, \ldots\}$), and use densely labeled datasets ($\mathcal{T} = \{1, 2, 3, \ldots\}$) for evaluation. See the Appendix for details.

After training to minimize NLL, we optionally refine our MDPFs to minimize the *mean-squared-error* (MSE) of the weighted particle mean, and report the induced Root-MSE (RMSE). Baseline methods do not estimate continuous posteriors, and thus Eq. (18) cannot be used for training. Instead we train and evaluate these methods with MSE and RMSE, before freezing the dynamics and measurement models. Then we construct KDEs, and optimize a bandwidth to report NLL values for comparison.

**Bearings-Only Tracking.** In the bearings only tracking task, we track the state of a variable-velocity car using noisy bearing observations from a radar station to the car generated as

$$y_t \sim \alpha \cdot \text{Uniform}(-\pi, \pi) + (1 - \alpha) \cdot \text{VonMises}(\psi(x_t), \kappa),$$

where $\psi(x_t)$ is the true bearing and $\alpha = 0.15$. Actions are not available. We use 5000 training and 1000 validation trajectories of length $T = 17$, and 5000 evaluation trajectories of length $T = 150$.

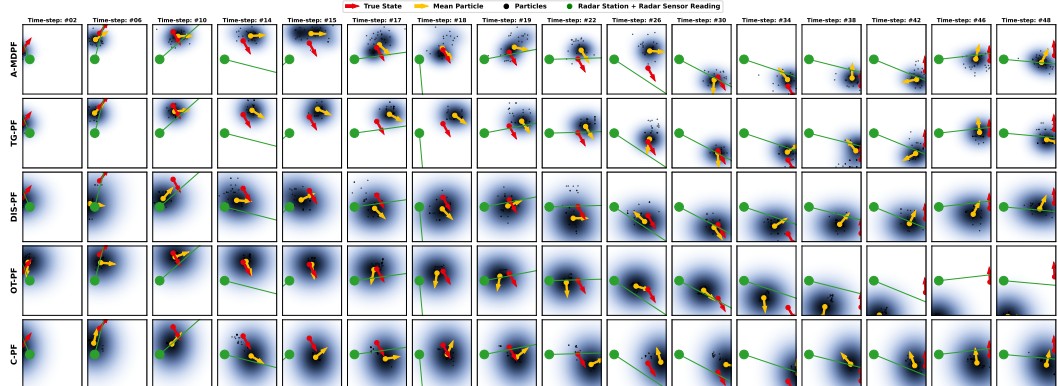

Figure 6: Example trajectories from the Bearings-Only tracking task. The radar station (green circle) produces bearing observations (green line) from the radar station to the car (red arrow). Note that observations are noisy and occasionally totally incorrect (see $t = 18$). We plot the posterior distribution of the current state (blue cloud), the mean particle (yellow arrow), and the $N = 25$ particles (black dots). A-MDPF is able to capture multiple modes in the state posterior (see $t = 15, 17, 18, 19$) while successfully tracking the true state of the car. Other models (DIS-PF, OT-PF, C-PF) spread their posterior distribution due to poor tracking of the true state.

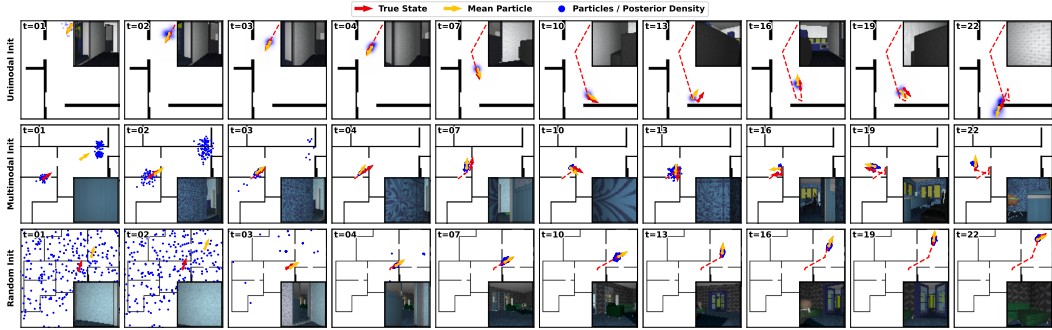

Figure 7: Example trajectories (rows), with observations in corner, from the House3D task using A-MDPF. *Top:* Initialized with $N = 50$ particles set to the true state with moderate noise, the posterior (blue cloud) closely tracks the robot. *Middle:* Multi-modal initialization with $N = 150$ particles (blue dots). *Bottom:* Naive initialization with $N = 1000$ particles drawn uniformly. A-MDPF maintains multiple modes until enough observations have been seen to disambiguate the true state.

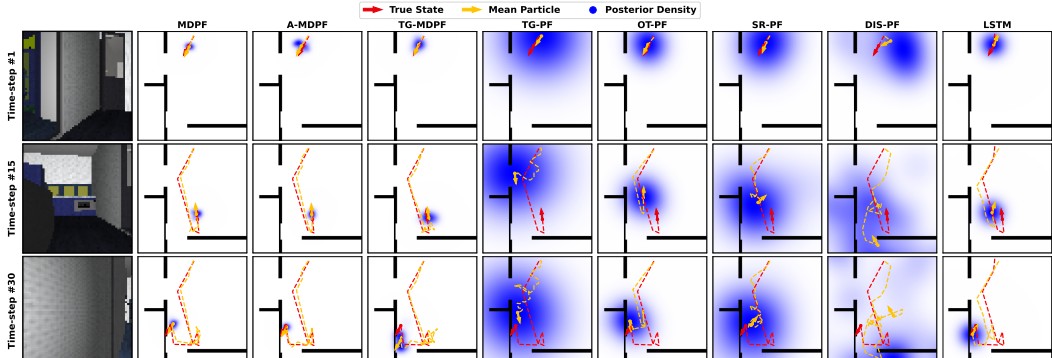

Figure 8: Example posteriors for three time steps (rows) of House3D tracking, for several discriminative PFs given the same low-noise initialization. We show the current true state and state history (red arrow and dashes), the posterior distribution of the current state (blue cloud, with darker being higher probability), and the estimated mean state and history (yellow arrow and dashes). Several baselines completely fail to track the robot.

Each method is evaluated and trained 11 times using $N = 25$ particles. To further highlight the stability of our IWSG estimator, we also train each method (except for IRG-MDPF) with the Epanechnikov kernel replacing the Gaussian in the KDE. Fig. 5 shows statistics of performance across 11 training runs. The very poor performance of IRG-MDPF is due to unstable gradients; without aggressive gradient clipping, IRG-MDPF fails to optimize at all. The benefits of IWSG are highlighted by the inferior performance of TG-MDPF. Stable multi-time-step gradients clearly benefit learning and are enabled by IWSG. Qualitative results are shown in Fig. 6.

**Deepmind-Maze Tracking.** In the Deepmind-Maze tracking task, adapted from [5], we wish to track the 3D state of a robot as it moves through modified versions of the maze environments from Deepmind Lab [48] using noisy odometry as actions, and images (from the robot's perspective) as observations. We alter the experimental setup of [5] by increasing the noise of the actions by five times, and using $N = 25$ particles for training and evaluation. We train and evaluate on trajectories from the 3 unique mazes separately, using 5000 trajectories of length $T = 17$ from each maze for training, and 1000 trajectories of length $T = 99$ for evaluation. We show results over multiple training and evaluation runs in Fig. 5, and include qualitative results in the Appendix.

**House3D Tracking.** This 3D state tracking task is adapted from [7], where a robot navigates single-level apartment environments from the SUNCG dataset [49] using the House3D simulator [50], with noisy odometry as action and RGB images as observations (see Fig. 7). Floor plans are input into measurement model via a modern Spatial Transformation Network architecture [44]. Training data consists of 74800 trajectories (length $T = 24$) from 199 unique environments, with evaluation data being 820 trajectories (length $T = 100$) from 47 previously unseen floorplans. We train and evaluate with $N = 50$ particles.

Table 1: Mean $\pm$ inter-quartile of evaluation metrics for all evaluation trajectories from the House3D task.

| Method | NLL | RMSE |
|---|---|---|
| LSTM | $12.54 \pm 3.43$ | $48.51 \pm 27.16$ |
| TG-PF | $15.50 \pm 2.21$ | $206.32 \pm 109.44$ |
| OT-PF | $14.87 \pm 2.31$ | $159.57 \pm 109.65$ |
| SR-PF | $14.92 \pm 2.57$ | $191.32 \pm 129.28$ |
| DIS-PF | $15.06 \pm 2.02$ | $212.58 \pm 133.69$ |
| TG-MDPF | $8.38 \pm 2.53$ | $35.77 \pm 16.67$ |
| MDPF | $8.18 \pm 2.72$ | $30.59 \pm 12.98$ |
| A-MDPF | $\mathbf{8.09 \pm 3.49}$ | $\mathbf{30.51 \pm 12.32}$ |

Due to the larger computational demands of House3D, we train each method once, reporting results in Table 1. Interestingly we find that TG-PF, OT-PF, SR-PF, and DIS-PF are unable to learn useful dynamics or measurement models. The LSTM model achieves better performance, but looking deeper we discover that it ignores observations completely, and simply blindly propagates the estimated state using actions with good dynamics (i.e., dead-reckoning). This strategy is effective for noise-free actions, but in reality the actions are noisy and thus the LSTMs estimated state diverges from the true state after a moderate number of time-steps. In contrast, our MDPF methods are able to learn good dynamics and measurement models that generalizes well to the unseen environments.

We also investigate MDPFs capacity for multimodal tracking in Fig. 7, by initializing A-MDPF with particles spread throughout the state space. When initialized with random particles or with particles clustered around regions with similar observations, A-MDPF does not have sufficient information to collapse the posterior and thus maintains multiple posterior modes. As the robot moves, more evidence is collected, allowing A-MDPF to collapse the state posterior to fewer distinct modes, before eventually collapsing its estimate into one mode containing the true state. Note that such estimation of multiple posterior modes is impossible for standard RNNs like the LSTM.

**Limitations.** A known limitation of all PFs is there inability to scale to very high-dimensional states. Sparsity increases as the dimension grows, and thus more particles (and consequently computation) are required to maintain the expressiveness and accuracy of the state posterior. Our MDPF is not immune to this issue, but we conjecture that end-to-end training of discriminative PFs will allow particles to be more effectively allocated within large state spaces, scaling better than classic PFs.

## 7 Discussion

We have developed a novel importance-sampling estimator for gradients of samples drawn from a continuous mixture model. Our results highlight fundamental flaws in the application of (implicit) reparameterization gradients to any mixture model with multiple modes. Applying our gradient estimator to the resampling step of a regularized discriminative PF, we obtain a fully end-to-end differentiable MDPF that robustly learns accurate models across multiple training runs, and has much greater accuracy than the biased PFs proposed in prior work. While our experiments have focused on synthetic tracking problems in complex virtual environments, future applications of our MDPF to real-world vision and robotics data are very promising.

## Acknowledgements

We thank Alexander Ihler for helpful discussions, and Zain Farhat for his assistance with the LSTM baseline. This research supported in part by NSF Robust Intelligence Award No. IIS-1816365 and ONR Award No. N00014-23-1-2712.

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

# A  Parameterization of the Neural Dynamics and Measurement Models

In our implementation of MDPF and A-MDPF, we define the dynamics and measurement models as a set of small learnable neural networks, Fig. 2. The dynamics model uses two encoder networks which encode the particles and the actions into latent dimensions. The particle encoder is applied individually to each particle, allowing for parallelization. The encoded actions and particles are used as input, along with zero-mean unit variance Gaussian noise, into a residual network to produce new particles.

The measurement model is also comprised of several small neural networks. Particles and observations are encoded into latent representations before being used as input into a simple feed-forward network. Our method makes no restrictions on the network architectures allowing modern architectures to be used. For high dimensional observations, such as images, or for more complex problems a sophisticated architecture such as Convolutional Neural Networks [43] or Spatial Transformer Networks [44] can be implemented.

Since particle states are interpretable, they are often in a representation not suitable for neural networks. For example angles contain a discontinuities (at $0$ and $2\pi$) which hinders their direct use with neural networks. To address this, non-learned transforms are applied to the particles, converting them into representations that are better suited for neural networks. In the case of angles transforms convert angles to and from unit vectors:

$$T(\theta) = (\cos(\theta), \sin(\theta)) \qquad T^{-1}(u,v) = \text{atan2}(u,v) \tag{19}$$

# B  Synthetic Learning Problem Details

In section 4, a simple synthetic PF learning problem was presented to illustrate the smooth and stable gradient estimates produced by our Importance Weighted Samples Gradient (IWSG) estimator while also highlighting the instability of reparameterization methods when applied to mixture models. Here we give additional details about this learning problem.

In this learning problem we define several regularized PFs with linear Gaussian dynamics and bimodal observation likelihoods:

$$p(x_t \mid x_{t-1}, a_t) = \text{Norm}(Ax_{t-1} + Ba_t, \sigma^2) \tag{20}$$

$$p(y_t \mid x_t) = w_1\text{Norm}(C_1x_t + c_1, \gamma^2) + w_2\text{Norm}(C_2x_t + c_2, \gamma^2). \tag{21}$$

For each PF method, we set the dynamics and measurement models to the true linear Gaussian dynamics and bimodal observation with perturbed parameters with the goal of learning the true values of the parameters. In this problem we hold $\sigma$ and $\gamma$ fixed and only perturb and learn the other parameters: $A$, $B$, $C_1$, $C_2$, $c_1$, $c_2$, $w_1$ and $w_2$.

We train three variants of the regularized PF with differing gradient estimation procedures for resampling. IWSG-PF uses our IWSG estimator, IRG-PF uses the implicit reparameterization gradients estimator, and TG-PF does not use any gradient estimator for mixture resampling; instead it simply truncates gradients to zero at resampling steps. We compare the stability and convergence of these three methods when learning the parameter values. For this problem, we decouple the resampling and posterior bandwidth parameters as in our A-MDPF model. For resampling we set $\tilde{\beta} = 0.05$, and for the posterior we set $\beta = 0.5$.

We also train a mixture Kalman Filter (KF) [11, 40] with the same true dynamics and measurement models, also initializing with perturbed parameters. Since this model has linear dynamics with Gaussian noise, and each component of the observation distribution mixture is also linear and Gaussian, we can apply mixture KFs to compute the exact posterior as a mixture of Gaussians. Dynamics are applied to each Gaussian component of the mixture using the KF update equations [11], keeping the number of components the same. When applying the measurement model, we simply multiply the KF mixture with the observation likelihood mixture, increasing the number of mixture components in the KF posterior. The number of components grows exponentially with time, so the mixture KF is not a practical general inference algorithm, but the computation remains tractable for short sequences. Special care must be taken when computing the weights of each mixture component after applying the observation likelihoods; we adapt the approach of Frei and Künsch [51].

To make the learning problem identifiable and to enforce the constraints $w_1 + w_2 = 1$, $w_1 \geq 0$, $w_2 \geq 0$, we parameterize $w_1$ and $w_2$ within the PF and KF models via the logistic of a single learnable parameter $v$:

$$w_1 = \frac{1}{1 + \exp(v)}, \qquad w_2 = \frac{1}{1 + \exp(-v)}. \tag{22}$$

To learn the dynamics and measurement model parameters, we setup a synthetic training problem with a dataset of 1000 sequences, each of length 5 time-steps. Using stochastic gradient descent with batch size 64, we train to minimize the negative log-likelihood loss of the true state given the posterior distribution at only the final time-step. For PF models, we use kernel density estimation [19] to estimate a mixture distribution using the final weighted particle set. For the mixture KF, the posterior distribution is simply a mixture model and no further processing is needed to compute the loss.

Fig. 9 shows the convergence and gradient estimates for all of the learned parameters when using the different gradient estimators. IRG's unstable gradients prevents convergence of all parameters, while IWSG converges smoothly and more rapidly than truncated gradients. Further we see that for parameters $c_1$ and $c_2$, the TG-PF method does not converge to the true value. TG-PF truncates gradients at the resampling, resulting in biased gradients. When the loss is computed at the final time, only information gained from that final time-step is present when updating the parameters, resulting in biased updates.

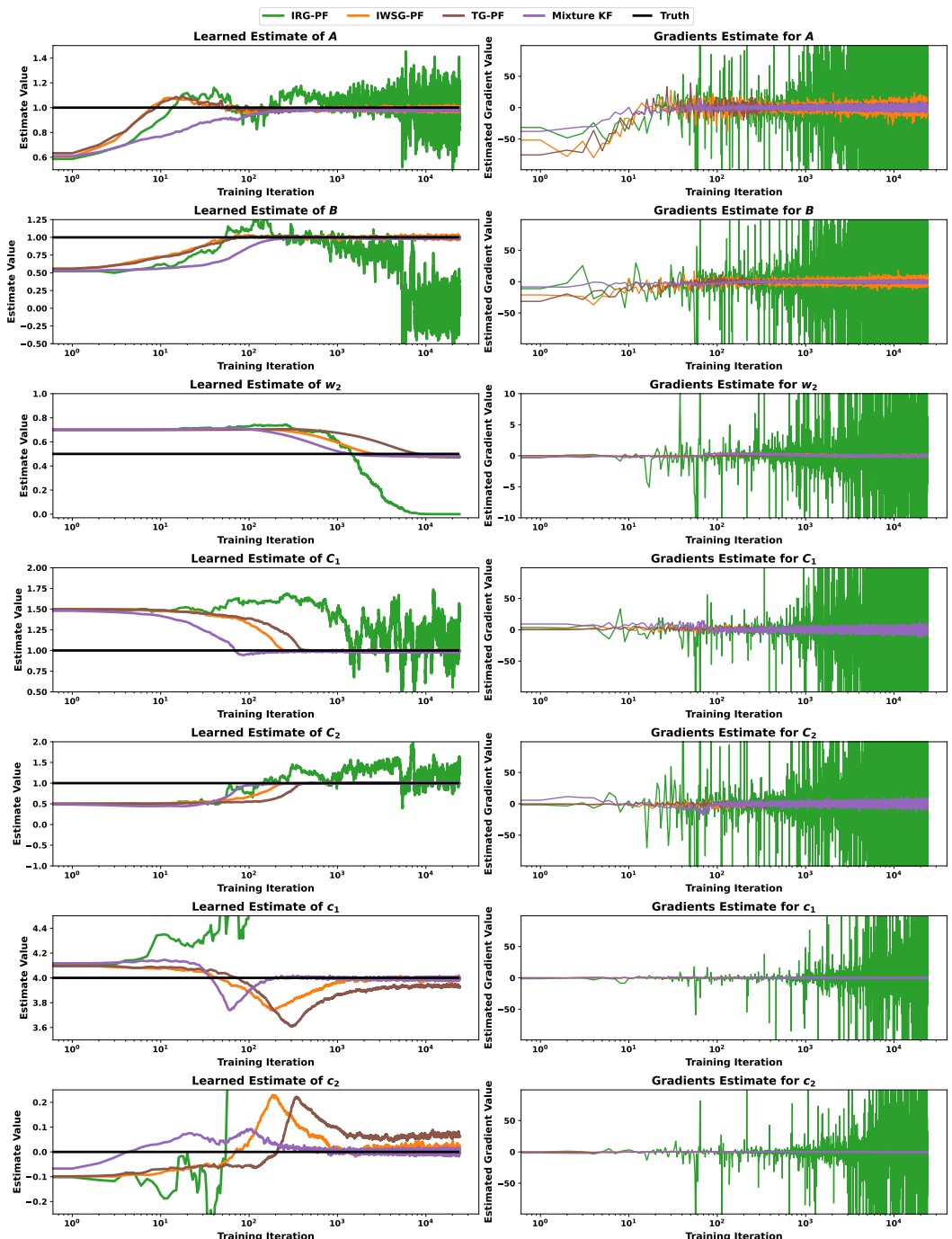

Figure 9: A simple temporal prediction problem where IRG has highly unstable gradient estimates. We use $N = 25$ particles when training IRG-PF, IWSG-PF, and truncated gradients (TG-PF). We show the learned values for all learnable parameters and their gradients during training. IRG produces unstable gradients which prevent parameters from converging at all, but IWSG allows for smooth convergence of all parameters. IWSG is faster than biased TG, and nearly as effective as (expensive, and for general models intractable) mixture KF. TG-PF fails to learn correct values for $c_1$ and $c_2$ due to biases from truncating gradients at resampling.

## B.1   Computation Requirements

The computational requirements for our MDPF and A-MDPF methods are quadratic with the number of particle during training, $O(N^2)$, due to gradient computation. During inference, the computation

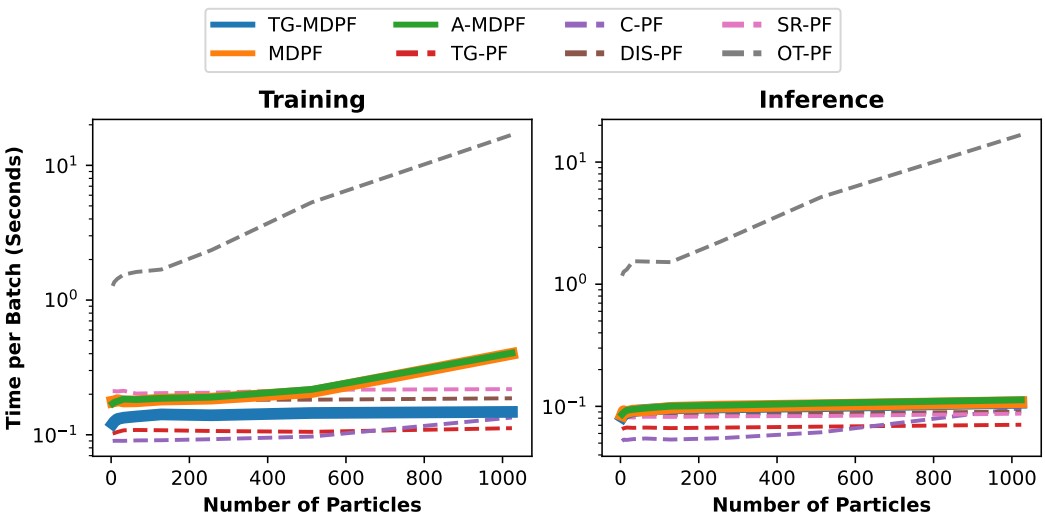

Figure 10: Computational time for various particle filters versus the number of particles on the Deepmind Maze task. A batch size of 24 is used. During training, the quadratic run-time of MDPF and A-MDPF is due to gradient computation. During inference MDPF and A-MDPF have linear run-time requirements. In contrast OT-PF must solve a complex optimal transport problem at both training and inference resulting in overall slow run-times.

requirement is linear in the number of particles, $O(N)$, since we mearly have to sample from the resampling mixture distribution rather than computing gradients for the samples. This is highlighted in fig. 10. Other methods except OT-PF are have computational complexity $O(N)$ during both inference and training. OT-PF requires solving an optimal transport problem with complexity $O(N^2)$ during both training and inference [8]. In practice we find that OT-PF has significant computational requirements resulting in slow run-times both at inference and testing.

## C   Additional Experiment Results

In this section we give additional qualitative experiment results for the various tracking tasks.

### C.1   Bearings Only Tracking Task

In fig. 11 we show additional qualitative results for various PF methods where it is apparent that A-MDPF and MDPF both track the true state well whereas other methods have wide posteriors due to their poor tracking ability. Figs. 12 13 14, 15, 16, 17 and 18 show additional trajectories for the various PF methods.

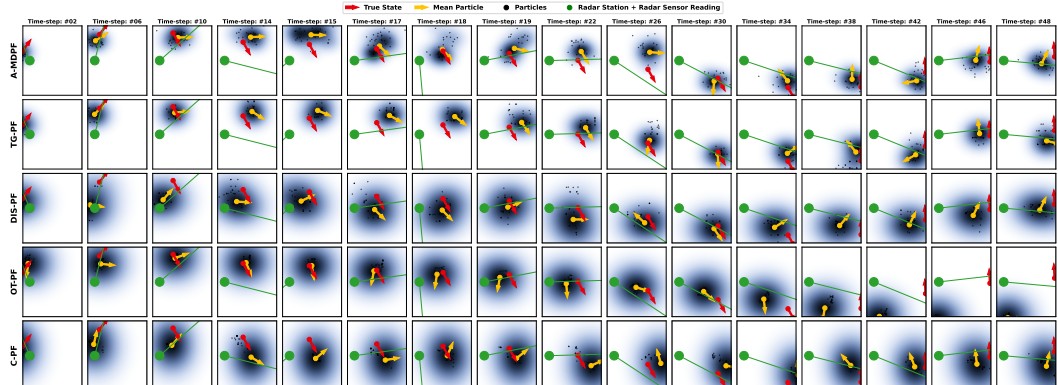

Figure 11: Example trajectory with various PF methods shown for the bearings only tracking task. The radar station (green circle) produces bearings observations (green line) from the radar station to the car (red arrow). Note that observations are noisy and occasionally totally incorrect. Shown is the posterior distribution of the current state (blue cloud, with darker being higher probability), the mean particle (yellow arrow) and the N = 25 particle set (black dots). A-MDPF and MDPF both have tight posterior distributions over the true state while maintaining tracking. Other methods such as DIS-PF, OT-PF and C-PF have wide posteriors showing their uncertainty due to their poor ability to track the true state.

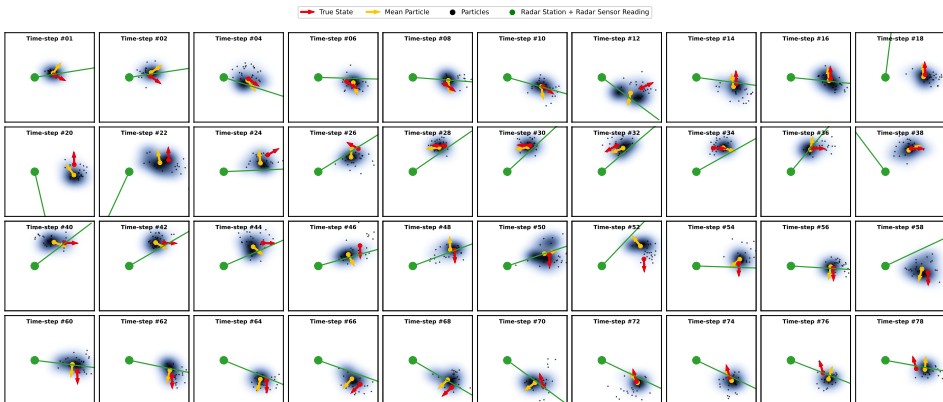

Figure 12: Example trajectory using A-MDPF shown for the bearings only tracking task. The radar station (green circle) produces noisy (and occasionally incorrect) bearings observations (green line) from the radar station to the car (red arrow). A-MDPF maintains a tight posterior distribution (blue cloud, with darker being higher probability) over the true state (red arrow) over time showing the effectiveness of our method. The mean particle (yellow arrow) and the N = 25 particle set (black dots) are also shown.

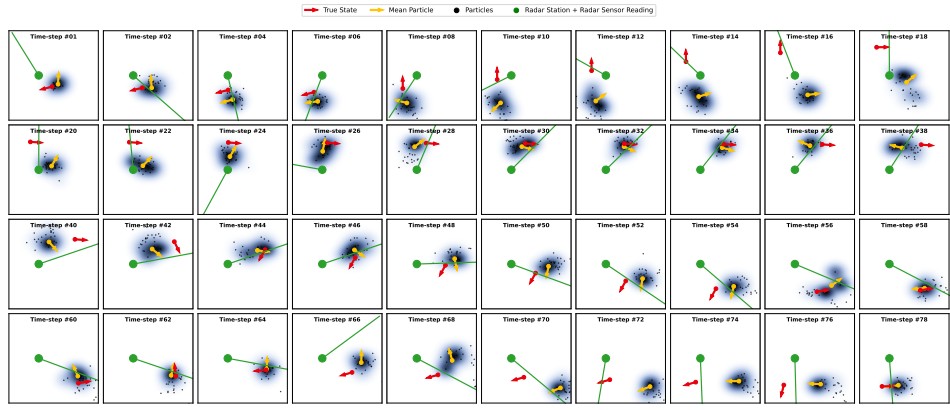

Figure 13: Another example trajectory using A-MDPF shown for the bearings only tracking task.

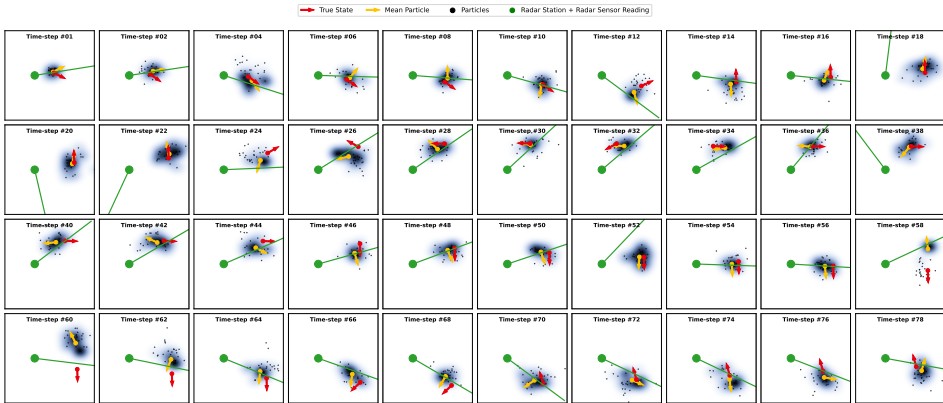

Figure 14: Example trajectory using MDPF shown for the bearings only tracking task. Similar to A-MDPF in figs. 12 and 13, MDPF accurately tracks the true state (red arrow) with a tight posterior density (blue cloud).

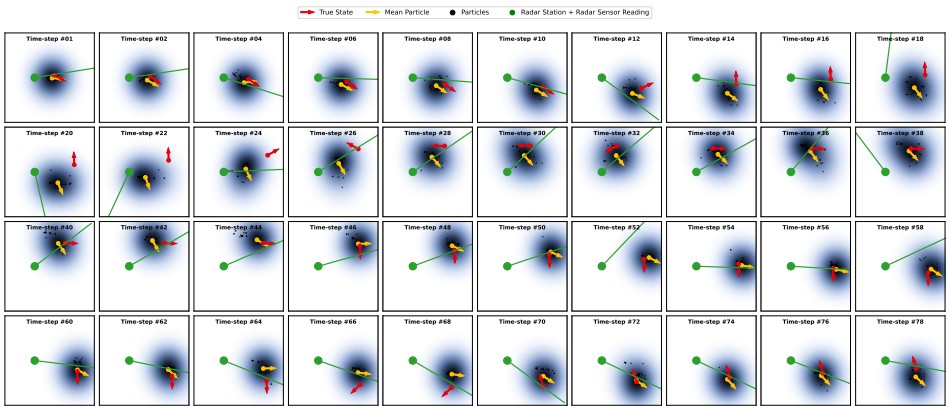

Figure 15: Example trajectory using DIS-PF shown for the bearings only tracking task. Due to the poorly learned dynamics and measurement models, DIS-PF often loses track of the true state (red arrow) as seen in time-steps $18 - 28$. This results in a large estimated posterior density (blue cloud) to account for the incorrectly placed particles (black dots) caused by poor dynamics and particle weighting.

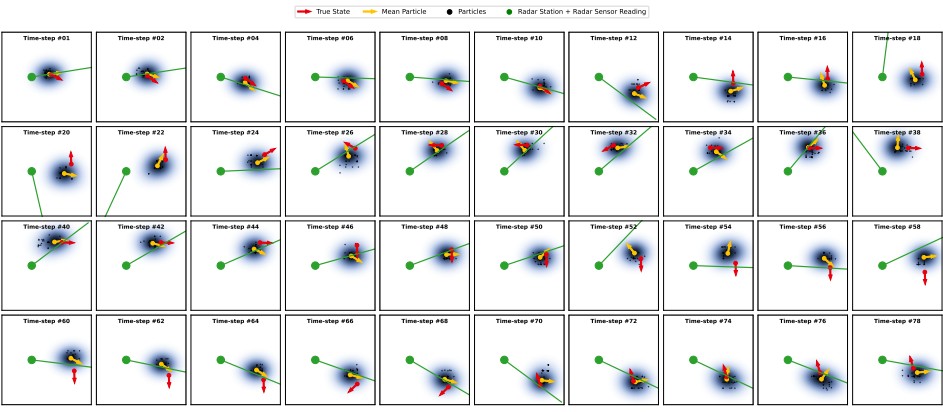

Figure 16: Example trajectory using SR-PF shown for the bearings only tracking task.

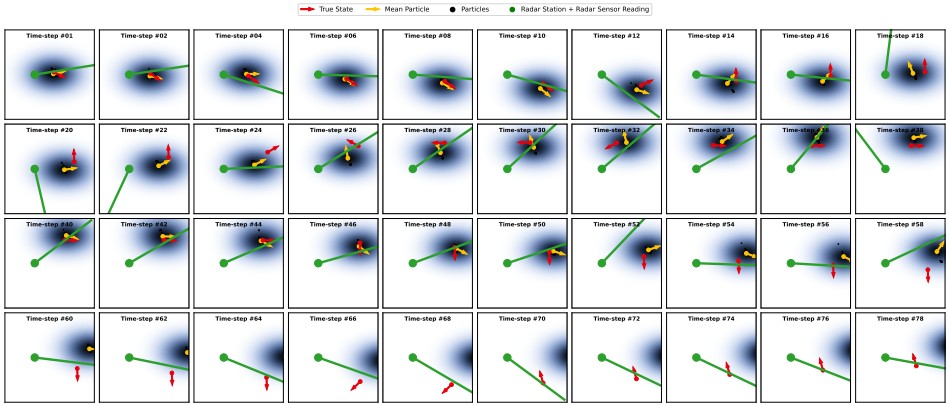

Figure 17: Example trajectory using OT-PF shown for the bearings only tracking task. Like DIS-PF in fig. 15, OT-PF has trouble tracking the true state (red) due to poorly learned dynamics and measurement models.

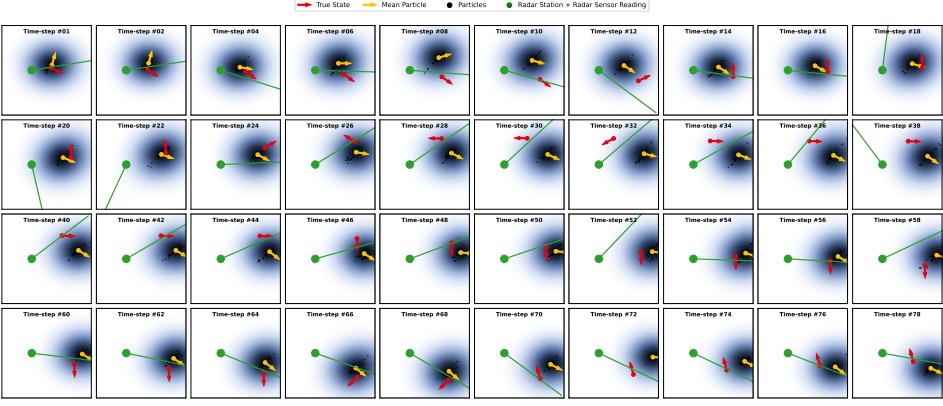

Figure 18: Example trajectory using C-PF shown for the bearings only tracking task. C-PF is unable to accurately track the true state (red) as shown by the relatively non-moving mean particle (yellow arrow).

## C.2 Deepmind Maze Tracking Task

In figs. 19, 20 and 21 we show additional qualitative results for various PF methods for mazes 1, 2 and 3, respectively. Again it is apparent that A-MDPF and MDPF both track the true state well whereas other methods have wide posteriors due to their poor tracking ability.

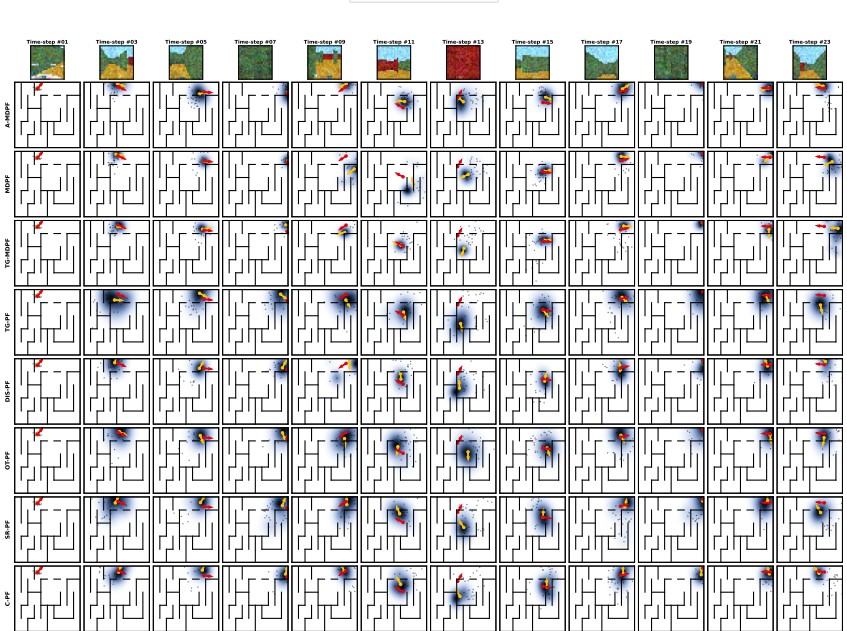

Figure 19: Example trajectory with various PF methods shown for the Deepmind maze tracking task for maze 1. Shown is the current true state (red arrow), the posterior estimate of the current state (blue cloud, with darker being higher probability), the mean particle (yellow arrow) and the N = 25 particle set (black dots).

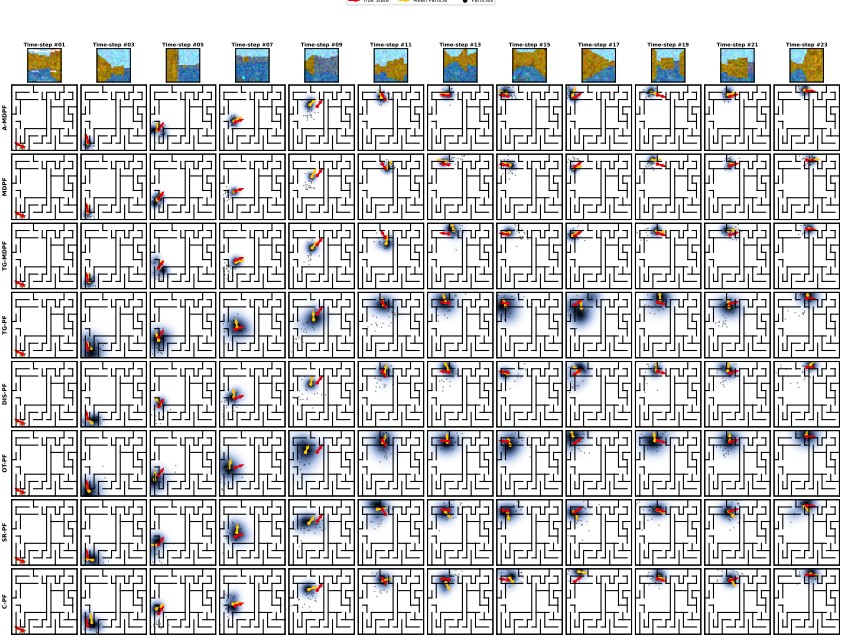

Figure 20: Example trajectory with various PF methods shown for the Deepmind maze tracking task for maze 2. Shown is the current true state (red arrow), the posterior estimate of the current state (blue cloud, with darker being higher probability), the mean particle (yellow arrow) and the N = 25 particle set (black dots).

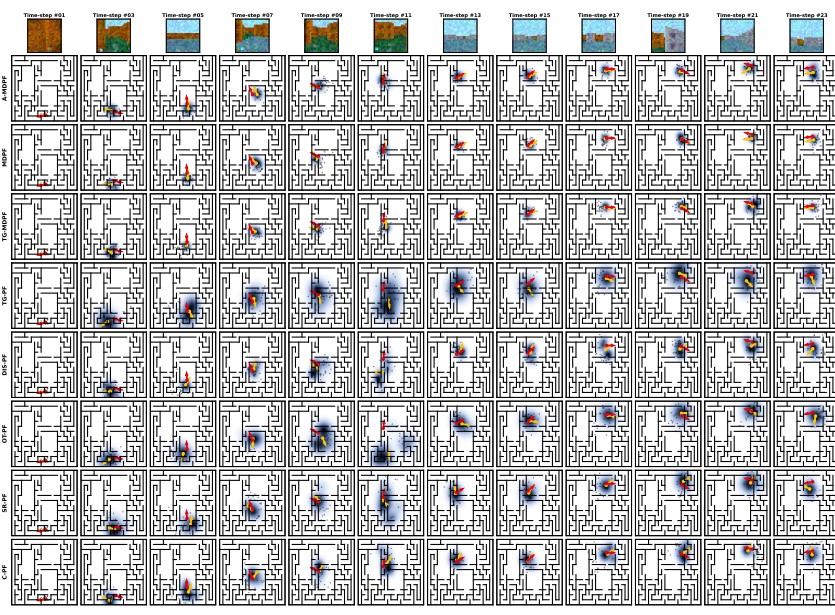

Figure 21: Example trajectory with various PF methods shown for the Deepmind maze tracking task for maze 3. Shown is the current true state (red arrow), the posterior estimate of the current state (blue cloud, with darker being higher probability), the mean particle (yellow arrow) and the N = 25 particle set (black dots).

## C.3 House3D Tracking Task

Figs. 22 and 23 show example trajectories for the House3D tracking task using A-MDPF and MDPF respectively. We see that our A-MDPF and MDPF methods both successfully track the true state over time as evident by the tight posterior over the true state as well as the closeness of the mean state to the true state. Fig. 24 shows an example trajectory using the LSTM model. It is clear that the LSTM model is ignoring observations and instead is blind propagating its state estimate using good dynamics and the actions (dead-reckoning). In later time-steps, the LSTM estimate drifts away from the true state however the predicted state trajectory looks similar to that of the true state, but with a growing offset, indicating blind propagation.

Fig. 25 shows qualitative results when using OT-PF. OT-PF is unable to learn good dynamics and measurement models and quickly diverges away from true state. Figs. 26, 27 and 28 give example trajectories for TG-PF, DIS-PF and SR-PF respectively. These methods fail to learn usable dynamics or measurement models with their trajectories essentially being random.

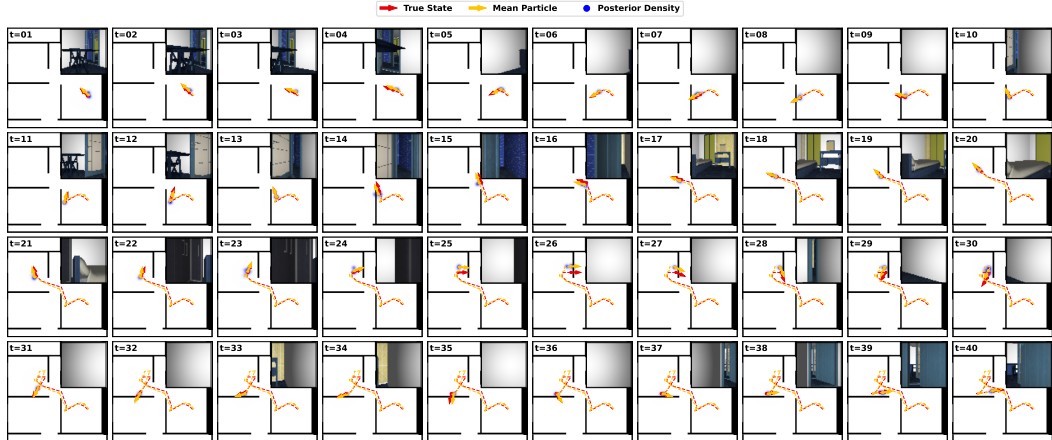

Figure 22: Example trajectory for the House3D tracking task using A-MDPF. The posterior density of the state is tight showing the confidence A-MDPF has in its estimate. The estimated posterior density (blue cloud, with darker being higher probability) also converges on the true state (red arrow) showing its accuracy. The mean particle (yellow arrow) closely tracks the true state. The observations are shown as inset images in the plots.

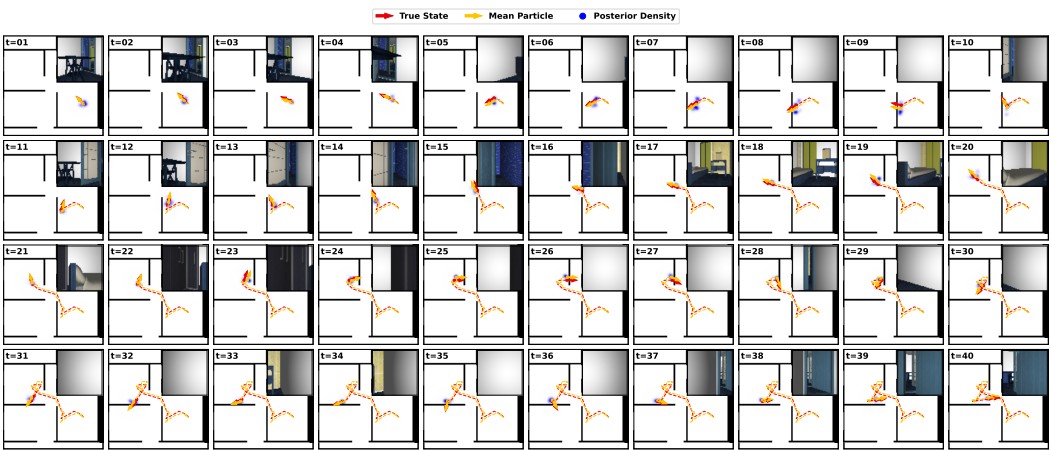

Figure 23: Example trajectory for the House3D tracking task using MDPF. Similar to fig. 22, MDPF accurately tracks the true state.

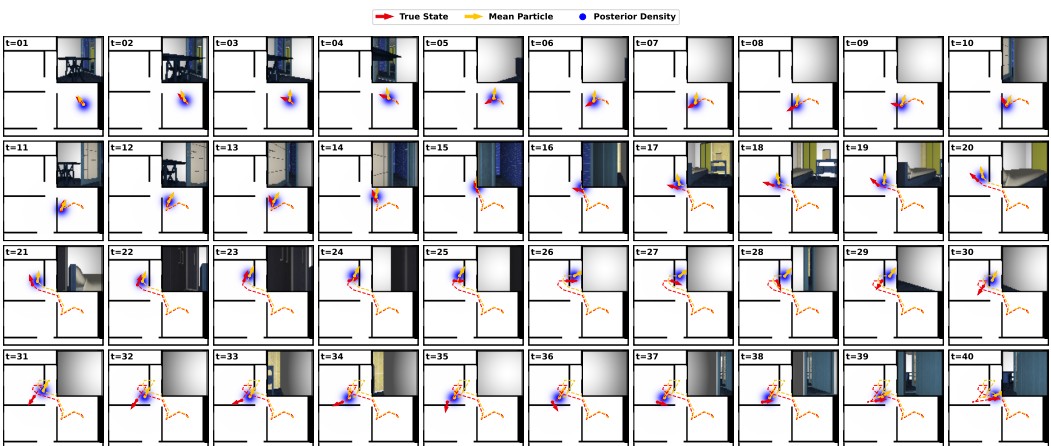

Figure 24: Example trajectory for the House3D tracking task using the LSTM model. The LSTM model does not us observations but rather blind-propagates the estimated state using the available noisy actions. This is seen by the true state (shown in red) and the estimated state (shown in yellow) having similar shapes but with increasing error.

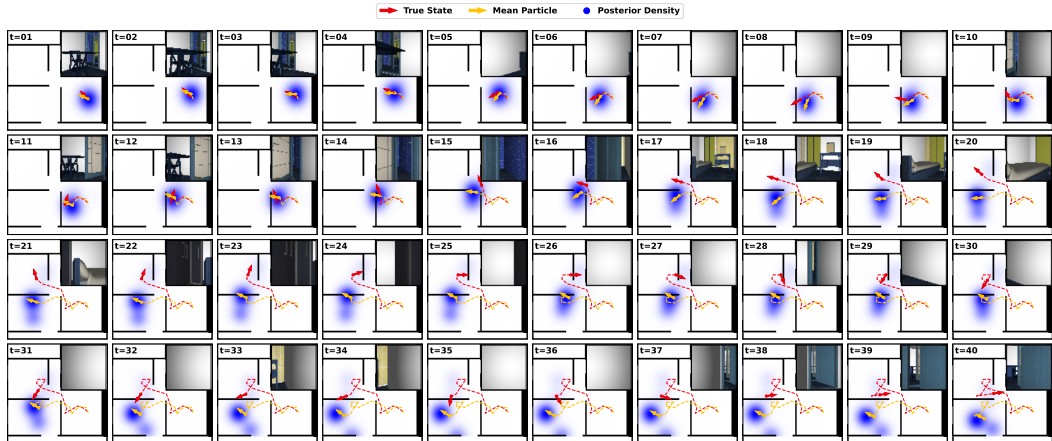

Figure 25: Example trajectory for the House3D tracking task using the OT-PF. OT-PF is unable to learn good dynamics or measurement models and thus the estimated posterior (blue cloud) diverges away from the true state (shown in red) after a couple time-steps.

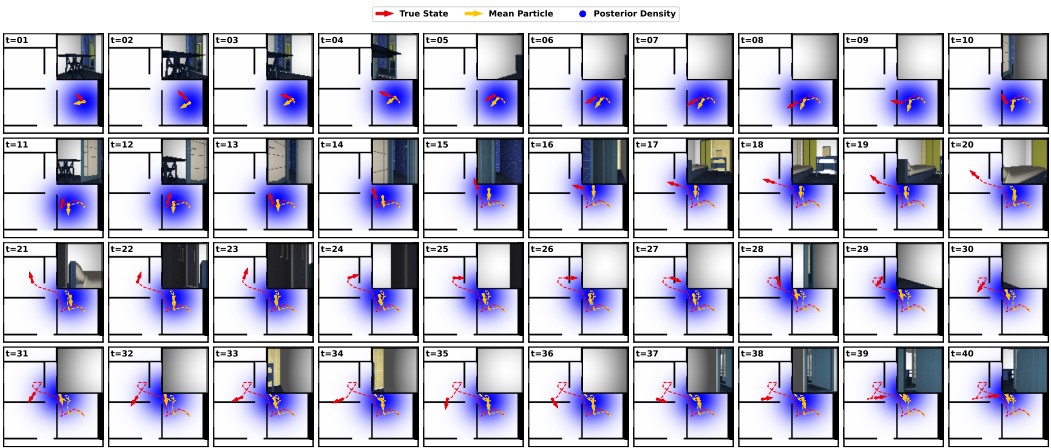

Figure 26: Example trajectory for the House3D tracking task using the TG-PF. TG-PF is unable to learn a usable dynamic or measurement model resulting in fairly random output. This is reflected in the large uncertainty of the estimated posterior (blue cloud) as well as the random nature of the mean state (shown in yellow).

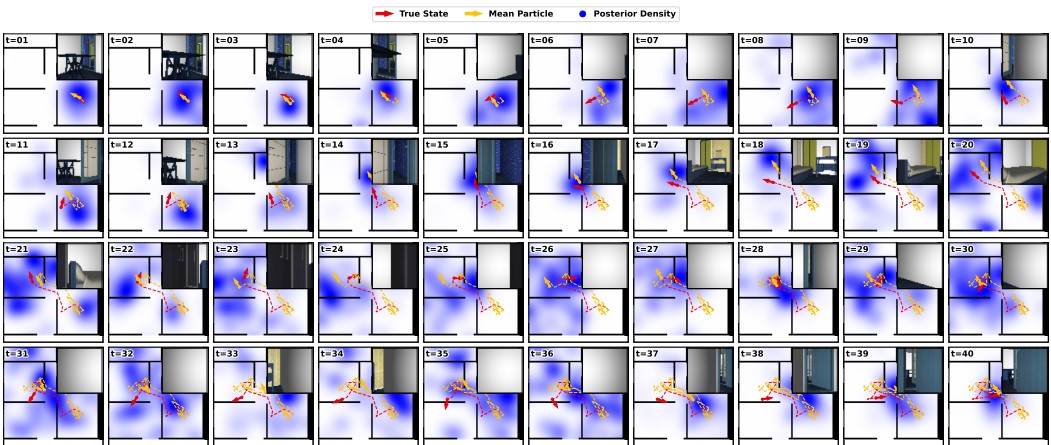

Figure 27: Example trajectory for the House3D tracking task using the DIS-PF. Similar to TG-PF in fig. 26, DIS-PF is unable to learn a usable dynamic or measurement model and therefore is unable to collapse the estimated posterior density (blue cloud) onto the true state (shown in red).

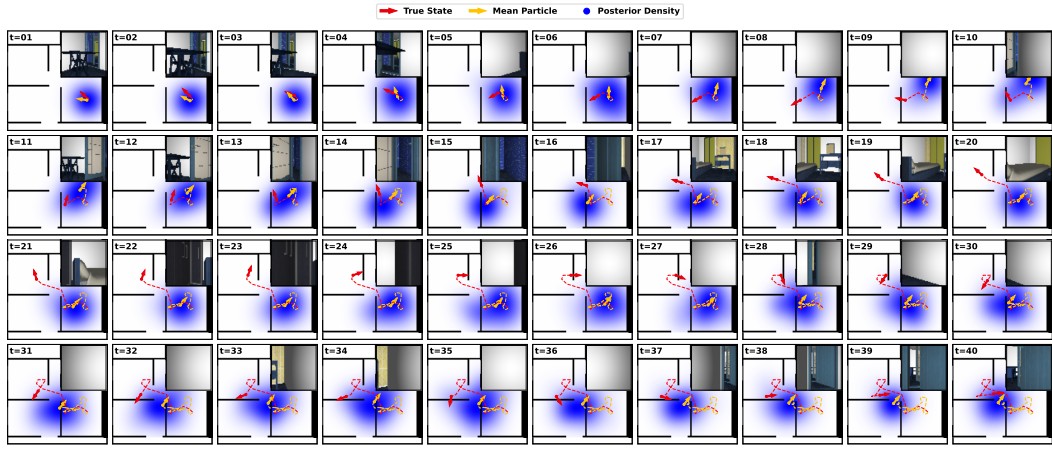

Figure 28: Example trajectory for the House3D tracking task using the SR-PF. Similar to TG-PF in fig. 26 and DIS-PF in fig. 27, SR-PF is unable to learn a usable dynamic or measurement model.

# D Experiment Details

In this section give additional details for the various experiments (tasks). We also provide specific neural architectures used in each task.

## D.1 Mean Squared Error Loss

When training the comparison PF methods, we use the Mean-Squared Error (MSE) loss. This is computed as the mean squared error between the sparse true states $s_t$ and the mean particle at each time $t$:

$$\mathcal{L}_{\text{MSE}} = \frac{1}{|\mathcal{T}|} \sum_{t \in \mathcal{T}} (x_t - \sum_{i=1}^{N} w_t^{(i)} x_t^{(i)})^2 \tag{23}$$

Here $\mathcal{T}$ are the indices of the labeled true states. During training, sparsely labeled true states are used and thus $\mathcal{T} = \{4, 8, 12, \ldots\}$. During evaluation we use dense labels, $\mathcal{T} = \{1, 2, 3, 4, \ldots\}$

For angular dimensions of $x_t^{(i)}$, we compute the mean particle by converting the angle into a unit vector, averaging the unit vectors (with normalization) and converting the mean unit vector back into an angle. Further we take special care when computing square differences for angles due to the discontinuity present at $0$ and $2\pi$.

## D.2 Pre-Training Measurement Model

We pre-train the measurement model for each task using the sparse true states. To pre-train, a random true state from the dataset is selected. A particle set is constructed by drawing samples from a distribution centered around the select true state and weights for this particle set are computed using the measurement model and the observation associated with the selected true state. A mixture distribution is then fit using KDE with a manually specified bandwidth. Negative Log-Likelihood loss of the true state is used as the training loss. We find pre-training not to be sensitive to choice of bandwidth so long as it is not too small.

The dynamics model cannot be pre-trained because of the lack of true state pairs due to the sparse true state training labels and thus must be trained for the first time within the full PF algorithm.

## D.3 Additional General Task and Training Details

Table 2: Hyper-parameters use for various PF methods.

| Parameter | Value |
|---|---|
| SR-PF [7] $\lambda$ | 0.1 |
| OT-PF [8] $\lambda$ (Bearings Only) | 0.5 |
| OT-PF [8] $\lambda$ (Deepmind) | 0.01 |
| OT-PF [8] $\lambda$ (House3D) | 0.01 |
| OT-PF [8] scaling | 0.9 |
| OT-PF [8] threshold | 1e-3 |
| OT-PF [8] max iterations | 500 |
| C-PF [31, 32] $\lambda$ | 0.5 |

For all tasks we wish to track the 3D (translation and angle) state of a robot, $s = (x, y, \theta)$, over time. This state space contains an angle dimension which is not suitable for neural networks. To address this issue, we convert particles into 4D using a transform $T(s) = (x, y, \sin(\theta), \cos(\theta))$ before using them as input into a neural network. Further we convert back into the original 3D representation to retrieve the a 3D particle after applying neural networks to the particles.

We use the Adam [52] optimizer with default parameters, tuning the learning rate for each task. We use a batch size of 64 during training and decrease the learning rate by a factor of 10 when the validation loss reaches a plateau. We use Truncated-Back-Propagation-Through-Time (T-BPTT) [47], truncating the gradients every 4 time-steps.

For all tasks we use the Gaussian distribution for the positional components of the state and the Von Mises distribution for the angular components when applying KDE. Here $\beta$ is a dimension-specific bandwidth corresponding to the two standard deviations for the Gaussians and a concentration for the Von Mises.

Some methods have non-learnable hyper-parameters that must be set. The values of the hyper-parameters are chosen via brief hyper-parameter search with the final chosen values shown in table 2. The hyper-parameter values are the same for all tasks unless otherwise specified. We find the regularization parameter $\lambda$ for OT-PF to be sensitive to the specific datasets.

We choose to make the dynamics position invariant. This prevents the learned PF models from memorizing the state space based on position but rather forces the learned dynamics to the true dynamics. Position invariance can be achieved by masking out the translational dimensions of the particle before input into the dynamics model.

### D.4 Bearings Only Tracking Task

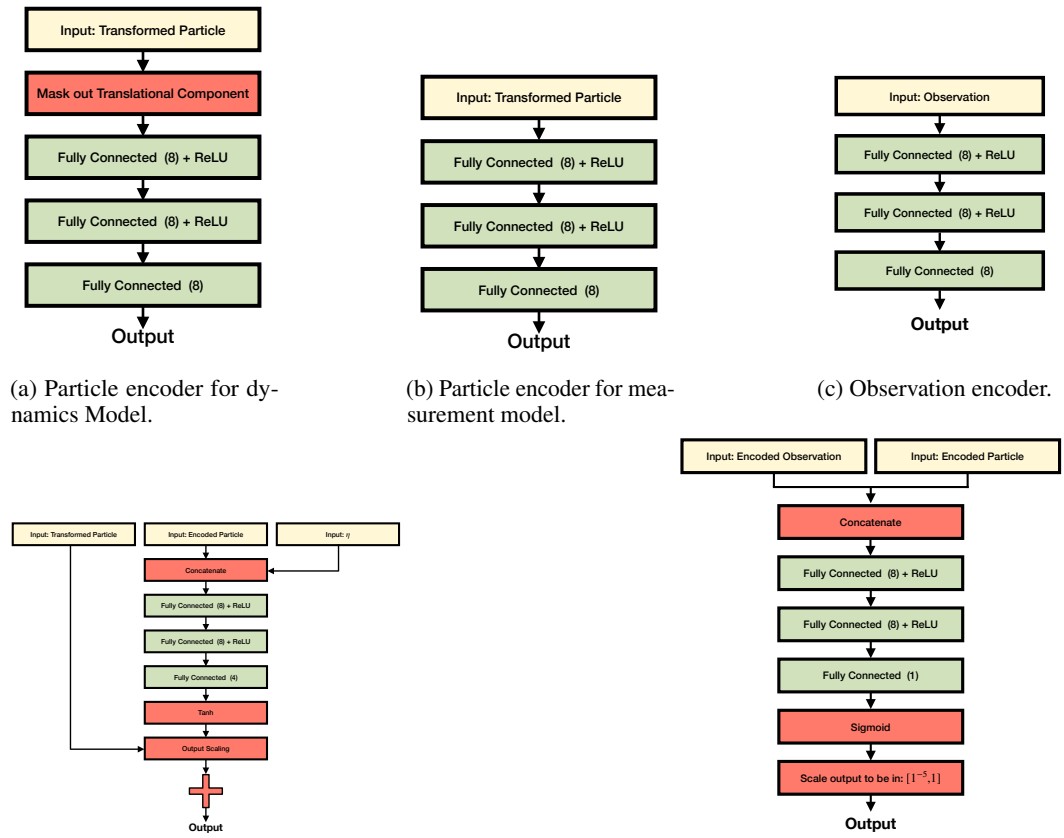

(a) Particle encoder for dynamics Model.

(b) Particle encoder for measurement model.

(c) Observation encoder.

(d) Dynamics model residual network, with actions omitted. Output scaling all dimensions to be within [-0.99, 0.99].

(e) Measurement model weights network.

Figure 29: Neural architecture for the networks within the dynamics and measurement models for the bearings only tracking task.

In this section we give additional task specific information and network architecture details for the bearings only tracking task.

### D.4.1 Additional Experiment Details

When training MDPF using the Negative Log-Likelihood (NLL) loss, we set the learning rate (LR) of the neural networks to be $0.0005$ and the bandwidth learning rate to $0.00005$. When training

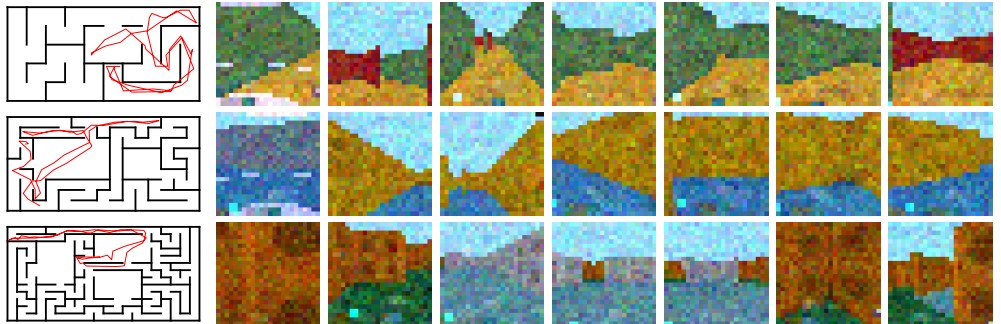

Figure 30: Example trajectories and observations from each of the three mazes from the Deepmind maze tracking task.

using MSE loss, we use 0.0001 and 0.00001 for the neural network and bandwidth learning rates respectively. For the discrete PF methods, we use LR = 0.0005 for training the neural network using the MSE loss. We use the same bandwidth for optimizing the bandwidth using the NLL loss when the neural networks are held fixed. For the LSTM [45], we use 0.001 as the learning rate for training the networks and LR = 0.0005 when optimizing the bandwidth parameter using NLL.

We apply gradient norm clipping, clipping the norm of the gradients to be $\leq 100.0$.

In addition to using the Gaussian distribution for KDE, we also train with the Epanechnikov distribution applied to the translational dimensions of the state. To do this we first train using Gaussian kernels before re-training with the Epanechnikov, using the already trained dynamics and measurement models as a starting point.

The observations $o_t$ are generated as noisy bearings from the radar station to the car:

$$o_t \sim \alpha \cdot \text{Uniform}(-\pi, \pi) + (1 - \alpha) \cdot \text{VonMises}(\psi_t, \kappa)$$

where $\psi_t$ is the true bearing, $\alpha = 0.15$ and $\kappa = 50$. Observations $o_t$ are angles and not suitable as input into a neural network. As such we convert the angular representation into a unit vector as is done with the angle dimensions of the particles.

### D.4.2 Network Architectures

Fig. 29 gives details on the neural network architectures used for the dynamics and measurement models within the PF for the bearings only tracking task. Since this task does not provide actions, the action encoder has been omitted.

### D.5 Deepmind Maze Tracking Task

In this section we give additional task specific information and network architecture details for the Deepmind maze tracking task.

### D.5.1 Additional Experiment Details

The Deepmind maze tracking task was first proposed in [5]. In this task, a robot moves through modified versions of the maze environments from Deepmind Lab [48]. Three distinct mazes are provided and we train and test on each maze individually. Example robot trajectories as well as example observations for each maze are shown in fig. 30. We modify this task by increasing the noise applied to the actions by 5 fold as well as using sparsely labeled true states during training.

When training MDPF using the Negative Log-Likelihood (NLL) loss, we set the learning rate (LR) of the neural networks to be 0.0005 and the bandwidth learning rate to 0.00005. When training using MSE loss, we use 0.0005 and 0.00005 for the neural network and bandwidth learning rates respectively. For the discrete PF methods, we use LR = 0.0005 for training the neural network using the MSE loss. We use the same bandwidth for optimizing the bandwidth using the NLL loss when the neural networks are held fixed. For the LSTM, we use 0.001 as the learning rate for training the networks and LR = 0.0005 when optimizing the bandwidth parameter using NLL.

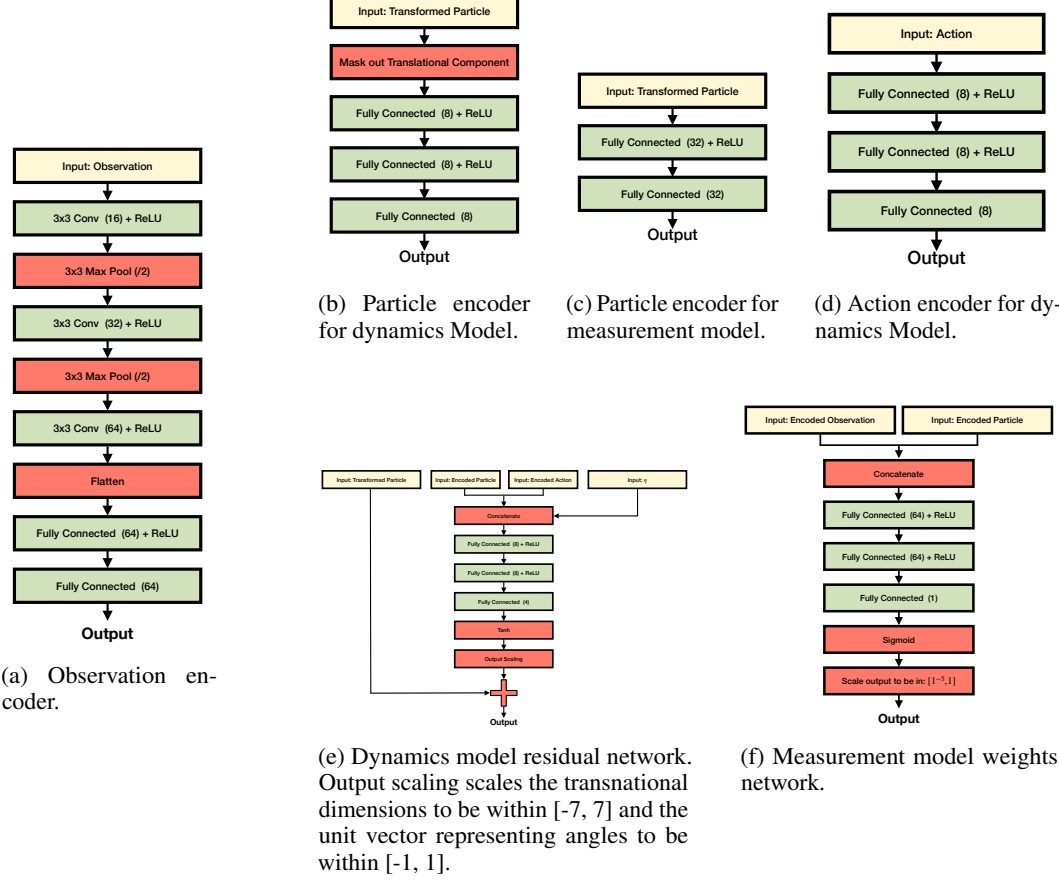

(a) Observation encoder.

(b) Particle encoder for dynamics Model.

(c) Particle encoder for measurement model.

(d) Action encoder for dynamics Model.

(e) Dynamics model residual network. Output scaling scales the transnational dimensions to be within [-7, 7] and the unit vector representing angles to be within [-1, 1].

(f) Measurement model weights network.

Figure 31: Neural architecture for the networks within the dynamics and measurement models for the Deepmind maze tracking task.

We apply gradient norm clipping, clipping the norm of the gradients to be $\leq 500.0$. Further we limit the bandwidth for the angle dimension such that $\beta \leq 4.0$. We do this only for this task as non-MDPF methods were prone to learning an unreasonably large bandwidth resulting in a near uniform distribution for the angular part of the state.

### D.5.2 Network Architectures

Fig. 31 gives details on the neural network architectures used for the dynamics and measurement models within the PF for the Deepmind maze tracking task.

### D.6 House3D Tracking Task

In this section we give additional task specific information and network architecture details for the House3D tracking task.

### D.6.1 Additional Experiment Details

When training MDPF using the Negative Log-Likelihood (NLL) loss, we set the learning rate (LR) of the neural networks to be $0.0005$ and the bandwidth learning rate to $0.00005$. When training using MSE loss, we use $0.0001$ and $0.00001$ for the neural network and bandwidth learning rates respectively. For the discrete PF methods, we use LR $= 0.0005$ for training the neural network using the MSE loss. We use the same bandwidth for optimizing the bandwidth using the NLL loss when the neural networks are held fixed. For the LSTM, we use $0.001$ as the learning rate for the LSTM layers and LR $= 0.0005$ when optimizing the bandwidth parameter using NLL.

We apply gradient norm clipping, clipping the norm of the gradients to be $\leq 250.0$ when training with NLL loss. When training with MSE loss, we disable gradient norm clipping.

### D.6.2 Network Architectures

Fig. 31 gives details on the neural network architectures used for the dynamics and measurement models within the PF for the House3D tracking task.

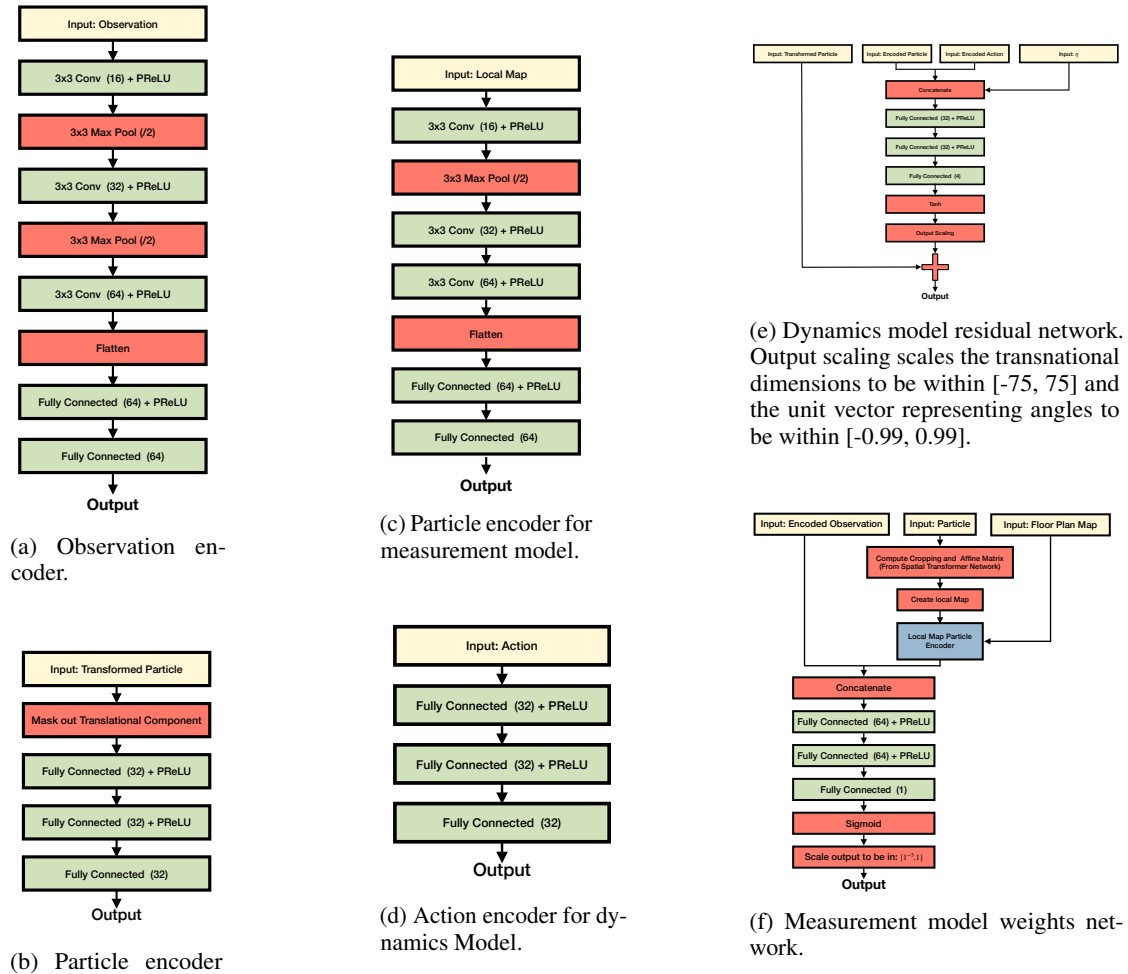

(a) Observation encoder.

(b) Particle encoder for dynamics Model.

(c) Particle encoder for measurement model.

(d) Action encoder for dynamics Model.

(e) Dynamics model residual network. Output scaling scales the transnational dimensions to be within [-75, 75] and the unit vector representing angles to be within [-0.99, 0.99].

(f) Measurement model weights network.

Figure 32: Neural architecture for the networks within the dynamics and measurement models for the House3D tracking task.

### D.7 LSTM Model Details

We train using LSTM [45] based recurrent neural network models for all tasks as a baseline comparison. For the LSTM, we use the same encoders as the PF methods. Fig. 33 shows the LSTM network architecture used for all tasks. We use the same output scaling the PF methods.

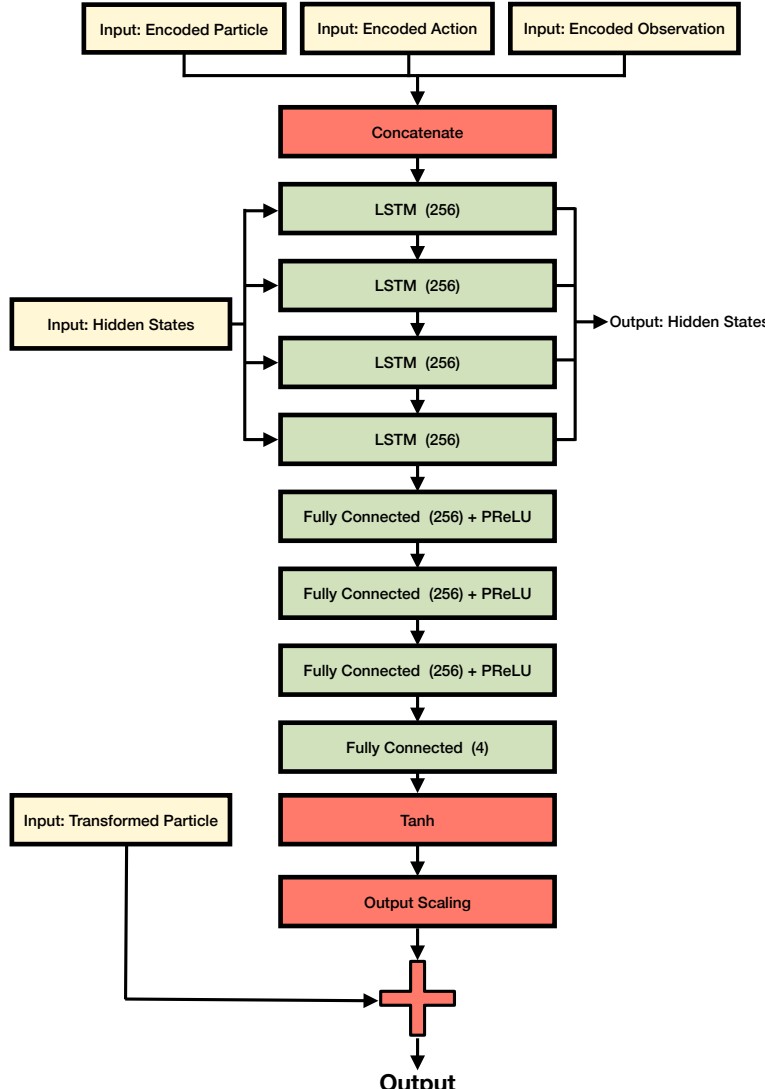

Figure 33: LSTM network architecture used for all tasks. The output is the 4D state which is then converted back into the 3D state using the inverse particle transform.

