# OpenReview forum: "Differentiable and Stable Long-Range Tracking of Multiple Posterior Modes"
_NeurIPS.cc/2023/Conference — NeurIPS 2023 poster_

### Official Review · Reviewer_FW5K · 2023-07-05

**Soundness:** 3 good
**Presentation:** 2 fair
**Contribution:** 2 fair
**Rating:** 5
**Confidence:** 3

**Summary:**

The paper proposes the "importance weighted samples gradient (IWSG)" estimator and describes its integration into a "mixture density particle filter (MDPF)" for state space architectures. Similar to regularized particle filters, the MDPF framework represents the continuous state posterior with a continuous mixture density, which is differentiable and can be trained end-to-end using the proposed IWSG. As a discriminative approach, the proposed particle filter does not require the specification of a generative model and, in contrast to other discriminative particle filters, returns unbiased and low-variance gradient estimates.

**Strengths:**

- Efficient and accurate particle-based estimation of complex posteriors in sequence models is an important topic and has a long history in the machine learning community.

- The paper includes an easy-to-follow recap of particle filtering and puts the proposed approach in context with many relevant discriminative particle filters, including a thorough discussion of their weaknesses that motivates both IWSG and MDPF.

- The proposed discriminative particle filter tackles a number of well-known challenges, such as differentiation through the resampling step, biased or high-variance gradient estimates, and limited expressiveness of the posterior parameterization.

- The experiments compare the proposed particle filter to a broad spectrum of relevant baselines and demonstrate MDPFs superior performance in three state estimation tasks.

**Weaknesses:**

- One of my concerns with this paper is its confusing presentation. The text often jumps between motivation, technical background, and related work, making it difficult to follow the intended train of thought. This problem is further aggravated by the fact that there are multiple technical streams: generative vs. discriminative particle filters, unbiased/low-variance gradient estimation, and inference in mixture models. There are many balls in the air and the paper has a hard time telling a streamlined story. Unfortunately, these are not the only problems with the presentation: Section 4 mixes technical background, the paper’s first main contribution (IWSG), and an experiment; Section 5 mixes the paper’s second main contribution (MDPF) with related work and contains no mathematical description of the model.

- The figures are weak for multiple reasons: (1) the font size is much too small; (2) the main text and the figure captions cannot be read independently, because the figures contain critical information that is not explained anywhere in the text; (3) the lack of subfigures ((a),(b),…) makes it difficult to map the information in the captions to individual subplots; and (4) the figures are not placed on the page they are referenced. Figure 4 is especially problematic as it describes the core of the paper’s contribution but is impossible to understand without further context (which the text does not provide either).

- The paper’s weak presentation also makes it difficult to assess the two contributions (IWSG and MDPF): (1) l.170 mentions that the proposed IWSG estimator is a continuous variant of the discrete estimator used in [6], but the exact relationship between the two remains unclear. Are there non-trivial challenges that prevent a direct generalization of [6] to a continuous setting?; (2) l.89/l.209 mentions that the proposed MDPF is a variant of [14,15], but again the differences are not explained in enough detail. Are there non-trivial challenges that prevent a direct application of [14,15] in a discriminative setting?

- One of the paper’s main claims is MDPF’s ability to compute unbiased and low-variance gradients. Unfortunately, neither the technical sections nor the experiments provide any *direct* evidence supporting this claim. I would have liked to see either a technical analysis of the proposed estimator's properties or experiments that go beyond the evaluation of NLL/RMSE and at least give empirical insights along those lines.

Overall, I feel this paper would benefit from another round of polishing before publication, including a reworked presentation and better positioning and differentiation of its contributions.

**Questions:**

- Section 4: It is mentioned that the optimal transport methods described in [44,45] are compatible with Gaussian mixtures (l.154f), so I’m curious why they are not part of the evaluated methods?

- Section 6: The design of the architecture used in the experiments (Sections 6.1-6.3) is not clear enough. Are all experiments based on Figure 4? If so, what are the encoders and transformations?

- Section 6: One of the arguments against [5] is its use of truncated gradients (l.118f), however, l.244 mentions that the experiments use truncated BPTT. Doesn’t that bias the MDPF gradients as well and makes the arguments in favour of MDPF obsolete?

- Section 6: I’m curious how the number of particles affects the reported performance metrics, and especially if the observed trends remain the same if the number of particles is significantly increased or decreased. If the authors have additional results along those lines it would be interesting to see them.

**Limitations:**

- I commend the authors for including a paragraph on the limitations of the proposed particle filter, such as its supervised nature and computational complexity on high-dimensional data.

- The paper does not discuss any ethical concerns related to the proposed method.

---

> ### Author Rebuttal · Authors · 2023-08-08
>
> Thank you for your thorough comments and helpful feedback, which we will use in future revisions.
>
> We concede that the presentation in the paper can be improved, and based on feedback we intend to move some technical details that are present in the supplement into the main paper text (see global response).  In the meantime, we would like to point the reviewer to Appendix A of the supplement, which we believe addresses many of your concerns.  Specifically, Appendix A contains information on prior works which clarifies how our IWSG extends work from [6], as well as the differences and similarities between our MDPF and classical regularized particle filters. Furthermore, in Appendix A.4 we give additional details on the general architecture/parameterization of MDPF and A-MPDF (details on encoders and transformations), and in Appendix D we give specific architectural details for all neural network components in MDPF and A-MDPF.
>
> With regard to optimal transport methods [44, 45] mentioned in the main paper text but for which we do not compare against: both [44] and [45] proposed gradient estimators focused on static distributions, and thus cannot be immediately applied to our particle filter context.  The text citing these papers was unclear about this, we will revise.  (Aspects of this work are also specialized to Gaussian mixtures, and cannot handle the von Mises kernels used in our experiments.)  For this reason, we instead compare to the Entropy-Regularized Optimal-Transport [8] (OT-PF) method, which was an optimal transport method specifically engineered to integrate with particle filters.
>
> With regard to truncated Back-Propagation-Through-Time (BPTT) biasing MDPF, it is true that truncating gradients biases all methods (including MDPF) to reduce variance. The main contribution of MDPF is the ability to propagate gradients temporally across multiple timesteps (between the sparse truncation points) which baselines like TG-PF [5] are unable to do.  With MDPF the BPTT truncations can be selected to balance bias and variance, as well as the computational requirements of training (frequent truncation is computationally cheaper, but more biased). TG-PF forces truncation at every time-step, and thus bias cannot be reduced even if additional computation is available.
>
> Unfortunately we did not run additional experiments with varying numbers of particles, as training multiple baseline methods for multiple trials (see box plots in Fig. 5,7) is already computationally demanding. But in the 1-page rebuttal figure PDF (Fig. 1), we do compare computational cost of the various methods with varying numbers of particles.
>
> [5] Rico Jonschkowski, Divyam Rastogi, and Oliver Brock. Differentiable particle filters: End-to-end learning with algorithmic priors. Proceedings of Robotics: Science and Systems (RSS), 2018.
>
> [6] Adam Scibior, Vaden Masrani, and Frank Wood. Differentiable particle filtering without modifying the forward pass. International Conference on Probabilistic Programming (PROBPROG), 2021.
>
> [8] Adrien Corenflos, James Thornton, George Deligiannidis, and Arnaud Doucet. Differentiable particle filtering via entropy-regularized optimal transport. International Conference on Machine Learning (ICML), 2021.
>
> [44] Martin Jankowiak and Fritz Obermeyer. Pathwise derivatives beyond the reparameterization trick. International Conference on Machine Learning (ICML), 2018.
>
> [45] Martin Jankowiak and Theofanis Karaletsos. Pathwise derivatives for multivariate distributions. International Conference on Artificial Intelligence and Statistics (AISTATS), 2019.

---

> > ### Comment · Reviewer_FW5K · 2023-08-14
> >
> > I want to thank the authors for their thoughtful rebuttal, in particular regarding the paper’s contribution relative to [6] and the role of truncated BPTT in the experiments. I still feel it would be in the authors' best interest to let this paper go through another round of polishing, so that the interesting ideas are not obfuscated by the presentation. I have increased my rating to 5 in response to the clarifications in the rebuttal.

---

### Official Review · Reviewer_FRJy · 2023-07-05

**Soundness:** 3 good
**Presentation:** 3 good
**Contribution:** 2 fair
**Rating:** 5
**Confidence:** 5

**Summary:**

This paper discusses the limitations of traditional particle filters in representing multiple posterior modes and their applicability to high-dimensional observations like images. Instead, the authors propose a method that leverages training data to learn particle-based representations of uncertainty using deep neural network encoders. The authors address the issue of biased learning and heuristic relaxations by representing posteriors as continuous mixture densities, allowing for unbiased and low-variance gradient estimates.



**Strengths:**

- This paper discusses an interesting problem.
- It is easy to read and follow the train of thought.
- Literature review is sufficient to grasp the idea of the idea of the paper.


**Weaknesses:**

- As MDPF relies on DNN parameterizations for both dynamic and measurement models, this introduces significant complexity to the model which can make training and inference more computationally expensive.
- Additional complexity of the model due to decoupling for A-MDPF as it requires careful tuning for separate bandwidth parameters.
- While the paper mentions that bandwidth parameters can be learned through end-to-end training, optimizing this parameter efficiently can require careful tuning and thus is challenging.
- I am still a bit uncertain about the authors' claim regarding the use of smaller NNs and non-learned operations in constructing the dynamic and measurement models.
- Lack of comparison with alternative frameworks for tracking and robot localizations and mostly applying to synthetic data.
- The limitations of the simulation setup in evaluating the MDPF and A-MDPF algorithms can be identified as follows:

     * The state estimation tasks focus on a 3D state consisting of translation and angle components (s = (x, y, θ)). While this simplification allows for manageable simulations, it may not capture the full complexity of state estimation in more challenging real-world scenarios with higher-dimensional state spaces.

     * The evaluation compares the MDPF and A-MDPF algorithms with a few selected particle filter variants (TG-PF, SR-PF, OT-PF, DIS-PF, C-PF) and an LSTM model. While this provides some insight into the performance of the proposed methods, a more comprehensive comparison with a wider range of state estimation algorithms would provide a better understanding of their strengths and weaknesses.

     * The training data uses sparsely labeled true states every 4th time-step, which may not accurately represent the labeling constraints in real-world applications. Dense labeling is crucial for accurately assessing the performance of state estimation algorithms. Although the evaluation uses densely labeled datasets, the training process may not fully exploit the benefits of dense labeling.



**Questions:**

Questions regarding the proposed importance weighted samples gradient (IWSG) estimator:

1- How effective is the IWSG estimator in reducing variance compared to other gradient estimation techniques?

2- Are there scenarios or specific mixture distributions where the IWSG estimator may still suffer from high variance?

3- Have there been any empirical evaluations to quantify the variance reduction achieved by the IWSG estimator?

4- How does the choice of the proposal distribution, q(z), affect the accuracy and efficiency of the IWSG estimator?

5- What is the computational cost associated with using the IWSG estimator compared to other gradient estimation methods?

Questions regarding the Algorithms:

1- How does the complexity of the deep neural networks used in MDPF and A-MDPF impact the computational requirements for training and inference?

2- Are there any strategies or techniques to improve the computational efficiency of these models?

3- How does the performance of MDPF and A-MDPF depend on the quality, size, and representativeness of the training data?

4- What happens when there is limited or biased training data available?

5- Can the learned parameters and decisions made by MDPF and A-MDPF be interpreted and understood?

6- Are there any techniques or approaches to enhance the interpretability of these models?

7- How does the performance of MDPF and A-MDPF depend on the choice and optimization of the bandwidth parameter (β) for kernel density estimation?

8- Are there any alternative methods or strategies to optimize this parameter effectively?

9- How does the decoupling of mixtures used for particle resampling and posterior state estimation affect the performance and training of A-MDPF?

10- Are there specific scenarios or domains where A-MDPF offers significant advantages over MDPF?

11- How well do MDPF and A-MDPF generalize to different problem domains or scenarios?

12- Have these models been tested on real-world datasets, and how do they perform compared to alternative approaches in those contexts?


**Limitations:**

Please refer to the weakness and question.

---

> ### Author Rebuttal · Authors · 2023-08-08
>
> Thank you for your comments and feedback. We address various comments and reviewer questions below.
>
> We would like to highlight Appendix A.4 (supplement) which gives additional details about the use of smaller neural networks and non-learned operations when constructing the dynamic and measurement models, and how this structure is motivated by particle filter mathematics.
>
> With regard to the computational cost of parameterizing the dynamics and measurement models by deep neural networks, we first note that MDPF makes no restrictions on the parameterization of either model other than differentiability (see Sec. 3).  Hand-engineered parametric models with limited tunable parameters may be substituted for either model if deep neural networks are too computationally expensive, though the expressiveness and flexibility to match complex training data would be lost.
>
> With regard to the additional bandwidth parameters in our A-MDPF methods, we find that careful manual tuning of the bandwidth for MDPF and A-MDPF is not necessary.  During end-to-end training, the bandwidth can be optimized simultaneously along with the dynamics and measurement models; this approach is successful in all of our experiments.
>
>
> IWSG Questions:
>
> 1. IWSG offers much lower variance gradient estimates compared to other estimators like IRG. We highlight this in Sec. 4 (main text) and Appendix B (supplement).
> 2. The only regime where IWSG has been observed to have high-variance is when the number of particles is extremely small (less than 20).
> 3. We empirically show the variance reduction in Fig. 3 (main text) and Fig. 2 (supplement), where it is clear that our IWSG estimator is much lower variance than IRG.
> 4. We frame IWSG as an estimator for estimating gradients for samples drawn from some mixture distribution $m(z)$.  The proposal distribution is always the “current” mixture distribution $q(z) = m(z)$, before gradient-based perturbation of the mixture parameters.  Thus, unlike some applications of importance sampling, there is no need to manually choose a proposal distribution.
> 5. Please see our global response to all reviewers, and the additional plot in our 1-page rebuttal PDF (Fig. 1).
>
> Algorithm Questions:
>
> 1,2. The majority of the computation time of MDPF is neural network evaluation, and thus more computationally intensive networks will require more compute time. MDPF gives no constraints on neural network architecture (other than they must be differentiable), as stated in Sec. 3 (main text).  Thus, methods developed to increase computational efficiency of general neural networks can be incorporated into MDPF training.
>
> 3,4. The quality of the learned dynamics and measurement models is dependent on the quality available data.  Our experiments show that training may be effective even when observations are temporally sparse, but of course, biased or insufficient data can lead to biased or inaccurate models.
>
> 5,6. The latent space (particles, weights, bandwidths) of MDPF and A-MDPF are defined by the training data, and are thus interpretable with real world meaning as highlighted in sections 2.1 and 5 (main text).  The parameters of the neural networks are not directly interpretable, as is the case with most neural networks. MDPF does not place any restrictions (other than differentiability) on the models used, and therefore interpretable parameterizations could be used.
>
> 7,8. The performance of MDPF (and A-MDPF) is dependent on the choice of bandwidth parameter $\beta$ and thus we choose to set it as a learned parameter. Very-small bandwidths ($\beta << 0.001$) can lead to numeric instability in float32 systems, and very-large bandwidths lead to poor MDPF performance. Classical methods, with known drawbacks, for setting the bandwidth exist and are referenced in section 2.2.  We find that learning the bandwidth end-to-end is simple and effective.
>
> 9 . Decoupling posterior and resampling mixtures as done in A-MDPF gives superior performance as shown in the results in section 6.  Training A-MDPF requires additional computation (due to the additional networks) but a pre-trained MDPF can be used to initialize A-MDPF allowing for faster training.
>
> 10 . We find that in general A-MDPF offers performance increases over MDPF due to the flexibility provided by decoupling the posterior and resampling distributions.  This can be seen in our results section 6 (specifically fig. 7).
>
> 11 . Our MDPF and A-MDPF learn problem-specific dynamics and measurement models, and are easily applied to any domain where appropriate data is available.  Application to test data with altered dynamics/likelihoods, without first retraining, may lead to poor performance.
>
> 12 . Prior work in this area has mainly been evaluated on synthetic datasets with high-dimensional observations, and we take the same approach. We evaluate on the same datasets as methods we directly compare to [1-3].  The House3D data is fairly realistic and the 3D environments it is based on data taken from real-world homes, but the observation images are still rendered with computer graphics.  Given the strong performance of MDPF, we believe experimentation with richer real-world datasets is a promising area for future research.
>
>
> [1] Adam Scibior, Vaden Masrani, and Frank Wood. “Differentiable particle filtering without modifying the forward pass”. International Conference on Probabilistic Programming (PROBPROG), 2021
>
> [2] Peter Karkus, David Hsu, and Wee Sun Lee. “Particle filter networks with application to visual localization”. Conference on Robot Learning (CORL), 2018.
>
> [3] Adrien Corenflos, James Thornton, George Deligiannidis, and Arnaud Doucet. Differentiable particle filtering via entropy-regularized optimal transport. International Conference on Machine Learning (ICML), 2021.

---

> > ### Comment · Reviewer_FRJy · 2023-08-14
> >
> > I'd like to thank the authors for their responses and clarifications. While I maintain my appreciation for their work, I believe it would greatly enhance the manuscript if the authors consider refining their content to address some of the concerns raised by the reviewers. I, however, keep my score as is.

---

### Official Review · Reviewer_53pr · 2023-07-06

**Soundness:** 3 good
**Presentation:** 3 good
**Contribution:** 3 good
**Rating:** 6
**Confidence:** 2

**Summary:**

The paper studies the problem of sequential state estimation using discriminative particle filters. Particle filters (or SMC) methods are widely used in this setting owing to their flexibility. Recently, inspired by the success of deep learning enabled by end-to-end differentiable methods, there has been interest in imparting similar properties to particle filters for improving performance, in particular when modelling temporally extended systems. The authors first review some existing approaches for particle filtering with a focus on regularized particle filters as well approaches for making the non-differentiable resampling step differentiable. The authors highlight several drawbacks with the existing methods with differentiable resampling, including biased gradients, numerical instability and high variance gradient estimates. The proposed method is based on leveraging implicit reparameterization gradients coupled with an importance sampling scheme to obtain an unbiased gradient estimator for continuous mixture distributions. The implicit reparameterization avoids the inversion of the standardization function and the importance sampling scheme reduces the variance of the estimates. The approach avoid various pitfalls of previous attempts to make the resampling step differentiable. The authors leverage this estimator in their Mixture Density Particle Filter method, which is evaluated through a wide variety of experiments with thorough quantitative and qualitative analysis - achieving improved performance across tasks.

**Strengths:**

* Sequential Monte Carlo approaches are widely used across various domains. Improvements to the algorithm can have a broad impact. The importance-weighted gradient estimator proposed in the paper can improve the performance on a wide-variety of tasks, beyond those studied in the paper.
* The proposed approach is principled and novel, leveraging recent advances in implicit reparameterization along with classical insights from the sampling literature to improve the performance of regularized particle-filters.
* The resulting method MDPF is conceptually simple, and can be incorporated easily as an alternative to existing regularized PF approaches.
* The paper is quite well written with a nice review of existing approaches and a lot of useful details about the experiments. I do think the introduction can be improved a bit.

**Weaknesses:**

* A key weakness of the approach is scaling to higher dimensional problems. It is a well known problem that importance sampling based estimators can still have high variance induced by some bad samples with high weights. (This is already discussed in the paper)
* While I appreciate the thorough experiments in the paper one issue is that most of the experiments are on relatively low dimensional problems.
* Although the details of the experiments are covered fairly well, the absence of code could be a problem for reproducibility efforts.

**Questions:**

* What would be potential solutions to scaling the method for higher dimensional problems?
* Have you tried the approach on some standard SMC tasks [1]? Does the gradient estimator improve performance there?

[1] An introduction to Sequential Monte Carlo by Nicolas Chopin and Omiros Papaspiliopoulos. https://particles-sequential-monte-carlo-in-python.readthedocs.io/en/latest/


**Limitations:**

The major limitations of the method are already highlighted in the paper - namely scaling to higher dimensional problems and reliance on labelled data.

---

> ### Author Rebuttal · Authors · 2023-08-08
>
> Thank you for your praise and feedback. We are grateful for the careful review of our work and appreciate your highlights to the broad applicability of our IWSG estimator beyond particle filtering and the simplicity and effectiveness of our MDPF method. Our intention is to release the code publicly (on Github) after some code cleanup and documentation.
>
> Regarding the application of particle filters in higher dimensions, we first note that our experimental domains are chosen to allow direct comparison to prior work, and are challenging in spite of their moderate dimension.  Scaling particle filters to higher-dimensional states does generally require more particles, but the number of required particles is heavily dependent on the posterior uncertainty; latent spaces with high posterior entropy require more particles [2]. By making the dynamics and measurement models of the particle filter learnable and effectively training them via our MDPF, the posterior state uncertainty can be reduced, allowing for fewer particles to be used. We conjecture that learning high-quality dynamics and measurement models, instead of hand-crafting models that only coarsely approximate reality, will allow particle filters to more efficiently scale to higher-dimensional problems.  As indirect evidence for this, we note that several of the baseline papers we cite used much-larger particle sets at test than during training, to compensate for poorly training models (due to biased gradients).  In contrast, our MDPF is effective when the test particle set size is exactly matched to training.
>
> With regard to “standard” SMC tasks as in [1], these methods assume known dynamics and measurement models, and thus the gradient-estimation innovations that our MDPF focuses on are not needed. Our Bearings-Only Tracking Task is similar to the tracking problem shown in 2.4 of [1], but more challenging:  the latent state has an additional dimension, and the dynamics and measurement models must be learned from training data (rather than being fixed to the true process used to generate synthetic data).
>
> [1] Nicolas Chopin and Omiros Papaspiliopoulos. “An introduction to Sequential Monte Carlo”. Springer 2020. ISBN 978-3-030-47844-5
>
> [2] Dieter Fox. “KLD-Sampling: Adaptive Particle Filters”. Advances in Neural Information Processing Systems (NeurIPS), 2001.

---

> > ### Comment · Reviewer_53pr · 2023-08-15
> > **Response to author rebuttal**
> >
> > Thank you for the response (and apologies for my tardy response to the rebuttal). I appreciate the authors response to my concerns. I will maintain my score recommending acceptance.

---

### Official Review · Reviewer_wjUs · 2023-07-07

**Soundness:** 4 excellent
**Presentation:** 3 good
**Contribution:** 4 excellent
**Rating:** 7
**Confidence:** 4

**Summary:**

The paper consider the problem of particle filter (PF)-based state estimation in nonlinear models with unknown dynamics and (discriminative) observation models.  The key challenge they address is the typical inability of traditional gradient learning approaches, applied to PF, to deal with (backprop through) the non-differentiable particle resampling stage.   To do this, the authors propose a novel Importance Weighted Samples Gradient (IWSG) Estimator, based on a kernel-density representation of the resampling step.  The IWSG is fully differentiable but is also unbiased and, in practice, low variance, particularly compared to the previously proposed Implicit Reparameterization Gradients (IRG) estimator.  The authors conduct an extensive set of experiments on simulated data (synthetic + simulator object-in-a-maze tracking scenarios) to demonstrate the utility of their IWSG.  IWSG is shown to outperform the competing approaches.

**Strengths:**

+ Well-written paper, with sufficient details (in main paper + supplement) that clearly introduce the problem, describe prior PF-based approaches, and build the proposed Mixture Density Particle Filter (MDPF) using a novel IWSG framework, following the ideas of Scibior et al., but using a kernel density-based (mixture density) resampler.
+ Addresses an important problem in the application of particle filter-based state estimators/trackers for models with unknown (parametric) dynamics and observation models (discriminative)
+ Propose two variants of MDPF, a baseline with a single mixture density used for both particle resampling and state estimation, and an improved A-MDPF, which employs different mixtures for the two tasks. A-MDPF is generally demonstrated to be more effective at the added computational cost of having two mixtures.
+ Extensive and convincing experiments conducted in simulated settings (both simple synthetic models and more complex 2D and 3D based simulators, c.f., Deepmaze and House3D with image based observations).  Experiments demonstrate that IWSG leads to consistently more accurate state estimates compared to either prior works / baselines (LSTM, TG-PF, OT-PF, SR-PF, DIS-PF, C-PF, TG-MDPF, and IRG-MDPF), together with lower estimator variances.

**Weaknesses:**

- Main paper does not clearly build the connection between Scibior et al. and the proposed approach, but this is clarified in the rather extensive supplement.  The proposed approach can be seen as perhaps rather obvious, once the KDE mixture model is employed for the resampler, but it is nevertheless a fairly ingenious combination of MD + IWSG.
- The authors acknowledge that PF-based approaches are not effective for high-dimensional state spaces and/or when the number of particles is large.  However, they do not clearly investigate this dependency (all experiments are conducted on max 3D state spaces + bearing).

**Questions:**

- How does the complexity of learning/inference depend on N?  Do you have experimental evidence that demonstrates this?
- Fig. 3 (supplement) shows some increasing variance trends in both gradients and states for MDPF (and A-MDPF).  Can you comment on this / explain?  Was this tendency observed in experiments in the simulator data?

**Limitations:**

The authors has addressed limitations

---

> ### Author Rebuttal · Authors · 2023-08-08
>
> Thank you for the positive comments and constructive feedback.  We agree that some of the discussion of baseline methods in the main text is not sufficiently clear, and will shift material from Appendix A to the main paper to address this issue.
>
> Please see our global response to all reviewers for discussion of computational complexity, and the new plot in Figure 1 of the rebuttal pdf.  While the computational cost of MDPF gradient computation with N particles scales as O(N^2) during training, the cost of MDPF resampling at inference/test time is O(N log(N)), comparable or better than baseline methods.
>
> With regard Fig. 3 (main paper) and Fig. 2 (supplement), the perceived increasing variance trends for both states and gradients of MDPF is a rendering artifact from using log scale on the horizontal axis.  We have re-rendered the plot with linear scale (shown in the 1-page rebuttal figure PDF, figure 2) to show that the variance of our MDPF does not tend to increase.

---

> > ### Comment · Reviewer_wjUs · 2023-08-17
> >
> > I am happy with the authors' rebuttal and will keep my score.

---

### Official Review · Reviewer_fwhN · 2023-07-07

**Soundness:** 3 good
**Presentation:** 2 fair
**Contribution:** 2 fair
**Rating:** 5
**Confidence:** 4

**Summary:**

This paper proposes an unbiased and low-variance gradient estimator for differentiable particle filters, where resampling steps incurs discrete changes that are non-differentiable in previous methods. The proposed method solved the problem by representing posteriors as continuous mixture densities, which is similar to the idea of regularized particle filters.

**Strengths:**

The paper is well-presented, motivation and proposed method are clearly explained, experimental results demonstrated the effectiveness of the proposed gradient estimator for differentiable particle filters, especially when the posterior is multimodal.

**Weaknesses:**

The performance of the proposed method when used in differnetiable particle filters trained with unsupervised loss when labelled data are not available is not discussed in the experiments/

**Questions:**

1. As far as I can see, all the experiments presented in the paper assumed the true latent state is available during training, do you have any idea about how to train differentiable particle filters when true states are not available in training but we still want to track the latent state in testing?
2. The paper claimed the proposed estimator is unbiased, but it is not obvious to me why it is unbiased and I expect to see a theoretical proof of this.

**Limitations:**

The proposed method is based on importance sampling, therefore some discussions on the variance of the estimator will improve the paper.

---

> ### Author Rebuttal · Authors · 2023-08-08
>
> Thank you for your comments and feedback.  To answer your specific questions:
>
> 1. Classical particle filters assume known dynamics and measurement models, and thus also implicitly assume the latent state has been manually defined; typically it has an interpretable real-world meaning such as the position or velocity of the system. Our MDPF extends the classical particle filter by allowing the dynamics and measurement models to be learned, training them end-to-end from (possibly sparse) observations.  In general to allow prediction of latent states, the state space must be defined somehow, either by hand-crafting of models (as in classical particle filters) or by training data (as in our paper).
>
> 2. Importance sampling is a broadly-applicable Monte Carlo method that, under very mild assumptions, produces unbiased estimates of expectations for any number of samples [1].  In general, importance sampling estimates the expectation with respect to some distribution $p(x)$ by drawing samples from some other distribution $q(x)$: $E_{p(x)}\Big[f(x)\Big] = \int_{x} p(x) f(x) dx = \int_{x}q(x) \frac{p(x)}{q(x)} f(x) dx =  E_{q(x)}\Big[\frac{p(x)}{q(x)}f(x)\Big]$. Our IWSG estimator is unbiased since it is derived from an unbiased importance-sampling estimator.  We will revise Sec. 4.2 to explain this more clearly.
>
> [1] C. P. Robert and G. Casella, “Monte Carlo Statistical Methods”, 2004.

---

> > ### Comment · Reviewer_fwhN · 2023-08-15
> > **Response  to author rebuttal**
> >
> > Thanks for the authors' response, I will keep my score as it is now and recommend an acceptance.

---

### Author Rebuttal · Authors · 2023-08-08

Thank you all for your feedback and helpful suggestions.  We want to address a few topics that were referenced by multiple reviewers.

First, for additional technical details about our methods as well as baselines, please refer to Appendix A of the supplement.  When drafting the paper, we put these details in Appendix A to allow more space for experimental details.  However, based on reviewer feedback, we now realize that this made it harder for readers to understand details of our mixture-density particle filter (MDPF) and gradient estimators.  In future revisions, we will move key parts of Appendix A back to the main paper (with space made available by shifting some parts of Sec. 6 to the appendix).

Several reviewers asked how the computational complexity of our MDPF scales with the number of particles N.  Figure 1 in the 1-page rebuttal PDF shows empirical trends for all methods. At training time, the complexity of MDPF gradient computation is O(N^2) per time step:  each of the N particles at the next time point could have been generated by any of the N mixture components at the previous time point, and the probabilities of these events must be computed.  Note that this overhead is not substantial for moderate numbers of particles, because for models of complex domains like images, resampling is much faster than neural-network likelihood evaluations whose cost remains O(N).  This extra computational cost is also key to avoiding the instability and poor performance of several baseline methods, which suffer from the “ancestor problem” (see Sec. 3).  Note also that this MDPF gradient computation is far cheaper than optimal-transport methods for aligning particles across time.

At inference or test time, gradient computation is not needed, and the MDPF has similar computational cost to “standard” particle filters.  Discrete resampling of N particles from the categorical distribution of particle weights requires O(N log(N)) time [1,2], and then particles may be perturbed by Gaussian noise (with learned bandwidth) in O(N) time.  This is in contrast with optimal-transport methods, which require expensive optimization at test as well as training.

We are currently in the process of cleaning and documenting our implementation of the MDPF, and will release code on Github when the paper becomes publicly available.

[1] https://en.wikipedia.org/wiki/Categorical_distribution#Sampling

[2] https://en.wikipedia.org/wiki/Alias_method

---

### Decision · Program_Chairs · 2023-09-21

**Decision:**

Accept (poster)

**Comment:**

This submission was reviewed by five reviewers in the field. The reviewer consensus reads that the paper is well-motivated and interesting, and the experimental results demonstrate the effectiveness of the proposed gradient estimator for differentiable particle filters. All reviewers recommend accepting this work. The concern by reviewer FW5K related to the presentation was well covered in the author reply. It would do the paper good to address the concerns, but I don't recommend a complete rewrite. Take the constructive feedback from reviewer FW5K into account when preparing the camera-ready version.